# In depth transcriptomic profiling defines a landscape of dysfunctional immune responses in patients with VEXAS syndrome

Hiroki Mizumaki[1,4], Shouguo Gao [1,4], Zhijie Wu [1], Fernanda Gutierrez-Rodrigues[1], Massimiliano Bissa [2], Xingmin Feng[1], Emma M. Groarke [1], Haoran Li [1], Lemlem Alemu[1], Diego Quinones Raffo[1], Ivana Darden[1], Sachiko Kajigaya[1], Peter C. Grayson[3], Genoveffa Franchini [2], Neal S. Young[1,5] & Bhavisha A. Patel[1,5]

VEXAS (Vacuoles, E1 enzyme, X-linked, Autoinflammatory, Somatic) syndrome is caused by inactivating somatic mutations in the *UBA1* gene. Here, we characterize the immunological landscape of VEXAS syndrome by performing multi-omics single-cell RNA analysis, cytokine multiplex assays, and in vitro functional assays on patients' peripheral blood. Our data reveals a broad immune system activation with upregulation of multiple inflammatory response pathways and proinflammatory cytokines. Unexpectedly, we find that monocytes have dysfunctional features irrespective of *UBA1* mutation status, exhibiting impaired efferocytosis and blunted cytokine production in vitro. In contrast, *UBA1*-mutated NK cells show an upregulation of the inflammation pathways and enhanced cytotoxicity. Within the lymphocyte subsets, predominantly *UBA1* wild-type, we identify clonal expansion of effector memory CD8+ T cells and skewed B cell differentiation with loss of transitional B cells and expansion of plasmablasts. Thus, our analysis indicates that VEXAS syndrome is characterized by profound alterations in both adaptive and innate immune systems, accounting for the complex pathophysiology of the disease, and provides a basis to understand the marked clinical heterogeneity and variable disease course.

VEXAS (Vacuoles, E1 enzyme, X-linked, Autoinflammatory and Somatic) syndrome is a newly described adult-onset autoinflammatory disease[1]. VEXAS is caused by inactivating somatic mutations in *UBA1*, an X-chromosome gene encoding ubiquitin-like modifier-activating enzyme 1 (UBA1) essential for cellular ubiquitylation in the ubiquitin-proteasome pathway. It is found in 1 of 4269 men older than 50 years[2].

In VEXAS, a reduction of the cytoplasmic UBA1 isoform by somatic *UBA1* mutations decreases the efficiency of endoplasmic reticulum (ER) associated protein degradation[1]. Imbalanced cellular proteostasis and accumulation of unfolded proteins cause excessive cytokine production and intrinsic autoinflammation. VEXAS patients have a wide range of multisystem inflammatory manifestations, including recurrent fever, relapsing polychondritis, vasculitis, pneumonitis, orbital inflammatory syndrome, and Sweet syndrome[3–6]. Hematological abnormalities resembling myelodysplastic syndrome, as well as plasma cell dyscrasia, and thromboembolic disease are also frequently observed[7–9].

[1]Hematology Branch, National Heart, Lung, and Blood Institute, National Institutes of Health, Bethesda, MD, USA. [2]Animal Models and Retroviral Vaccines Section, National Cancer Institute, National Institutes of Health, Bethesda, MD, USA. [3]Vasculitis Translational Research Program, National Institute of Arthritis and Musculoskeletal, and Skin Diseases, National Institutes of Health, Bethesda, MD, USA. [4]These authors contributed equally: Hiroki Mizumaki, Shouguo Gao. [5]These authors jointly supervised this work: Neal S. Young, Bhavisha A. Patel. ✉e-mail: zhijie.wu@nih.gov; youngns@nhlbi.nih.gov

The original VEXAS report described the upregulation of inflammatory pathways in neutrophils and monocytes and the activation of the unfolded protein response (UPR) pathway in myeloid cells[1]. We recently showed that *UBA1*-mutated (mt*UBA1*) bone marrow (BM) myeloid cells upregulate multiple inflammatory response pathways, as compared to wild-type *UBA1* (wt*UBA1*) cells, indicating that autoinflammation in VEXAS is initiated by mt*UBA1* hematopoietic stem and progenitor cells (HSPCs)[10].

In VEXAS, somatic *UBA1* mutations occur in the hematopoietic stem cells (HSC), but they are ultimately restricted to mature myeloid cells in peripheral blood (PB), such as neutrophils and monocytes. Conversely, mature T and B lymphocytes, and natural killer (NK) cells are rarely mt*UBA1*[11]. All these lymphoid cells are decreased in the PB of VEXAS patients, implying wt*UBA1* lymphoid cells may not be actively involved in the pathophysiology of VEXAS. However, plasma cell dyscrasias and monoclonal B cell lymphocytosis are observed in patients despite decreased B-lineage cells and the absence of *UBA1* mutations in those cells[7]. In addition, T cells appear to be clonally expanded in the BM[10].

In this study, we use single-cell multi-omics to comprehensively describe immunological features of VEXAS syndrome. In addition, we employ genotyping of transcriptomes (GoT) to link genetic variation to single-cell transcriptomes. Our result reveals a broad immune system activation, with upregulation of multiple inflammatory response pathways in disease. Dysfunctional transcriptional features are present in VEXAS monocytes, irrespective of their *UBA1* mutation status. mt*UBA1* NK cells present with an increased inflammation signature of enhanced cytotoxicity. Among lymphocyte subsets, predominantly wt*UBA1*, we identify clonal expansion of effector memory CD8⁺ T cells and skewed B cell differentiation, with loss of transitional B cells and expansion of plasmablasts. In VEXAS, systemic inflammation is driven by innate and adaptive immune dysregulation, including both mt*UBA1* and wt*UBA1* cells.

## Results

### Study design and single-cell RNA profiling of PB immune cells
From a VEXAS cohort of 56 patients prospectively evaluated at the NIH Clinical Center, nine patients (all males, median age 70 [range, 60–74]) and five age- and sex-matched healthy donors (all males, median age 66 [range, 61–70]) were studied by multi-omics single-cell RNA analysis (Fig. 1a). PB mononuclear cells (PBMC) isolated from these individuals were subjected to single-cell RNA sequencing (scRNA-seq) and single-cell T cell receptor/B cell receptor sequencing (scTCR/BCR-seq) using the 10x Genomics platform, followed by various bioinformatics analyses using high-quality immune cells obtained from VEXAS patients (125,806 cells) and healthy donors (52,362 cells). Details of patients' characteristics are provided in Supplementary Data 1. Consistent with previous reports[1,4,5], patients were cytopenic, with prominent macrocytic anemia, thrombocytopenia, monocytopenia, and lymphocytopenia (Fig. 1b).

Ten major cell subtypes were identified in VEXAS patients and healthy donors by Granulator (v1.12.0)[12] based on scRNA-seq gene expression profiles: CD4⁺ and CD8⁺ T cells, B cells, plasmablasts, natural killer (NK) cells, mucosal-associated invariant T cells (MAIT), gamma delta T cells (gdT), monocytes, neutrophils, and dendritic cells (DC) (Fig. 1c, d and Supplementary Fig. 1a, b). GoT libraries were constructed from cDNA samples with *UBA1* p.M41 mutations (n = 8; VEXAS 9-15,17) using a previously described method with some modifications[13]. Since GoT was employed to identify *UBA1* mutations within cDNA libraries from VEXAS patients and was not suitable for detection of splice site mutations, GoT library was not constructed from healthy donors or VEXAS 16, who has a *UBA1* mutation at a splicing site (c.118-1 G). While the cb_sniffer analysis[14] of the 10x gene expression data detected *UBA1* transcripts in only 4.9% cells, the GoT approach identified *UBA1* transcripts in 16.8% cells (19,542 out of

116,547 cells) (Fig. 1e and Supplementary Fig. 1c, d; Supplementary Data 2). mt*UBA1* transcripts were enriched in monocytes, DCs, and NK cells, but absent in B and T cells (Fig. 1f and Supplementary Fig. 1e; Supplementary Data 2), consistent with our previous single-cell DNA sequencing data of PB from VEXAS patients[11].

### Altered cellular profiles in VEXAS patients
PBMC composition in the scRNA-seq data of VEXAS patients was abnormal. Compared to healthy donors, differential abundance analysis showed that VEXAS patients had prominently increased plasmablasts and moderately higher CD8⁺ and CD4⁺ T cells, while NK cells, MAITs, DCs, and B cells were decreased (Fig. 2a, b and Supplementary Fig. 2a). Many genes related to the inflammatory response pathways were differentially upregulated in VEXAS, particularly in the IFN-α, IFN-γ, and TNF response pathways (Supplementary Fig. 2b). UMAP embedding of gene module scores showed that the IFN-α response pathway was uniformly upregulated in all major cell subtypes; the IFN-γ and TNF response pathways were predominantly upregulated in CD8⁺ T, NK cells, and monocytes in VEXAS patients (Fig. 2c). Gene module scores of the IFN-α, IFN-γ, and TNF response pathways were significantly higher in most cell subtypes of VEXAS than in cells from healthy donors (Fig. 2d).

Cell activation of different immune cell types in VEXAS was associated with increased upregulation of ligand-receptor interactions involved in inflammation and cell adhesion; ligand-receptor interactions included *IL18-IL18R1*, *IL1B-IL1R2*, and *THBS1-ITGB1*. Ligands and receptors were coordinately upregulated among innate immune cells (monocytes, DCs, and NK cells) and adaptive immune cells, such as CD8⁺ T and B cells (Fig. 2e).

Broad immune cell activation in VEXAS translated to increased pro-inflammatory cytokines in plasma of patients. As expected, IL-6, IL-18, IFN-γ, and TNF expression levels were higher in VEXAS than in healthy donors. Expression of chemokines including CXCL5, CXCL6, and MCP4 was lower in VEXAS (Fig. 2f and Supplementary Fig. 2c). By principal component analysis (PCA), there was clear separation of plasma cytokine expression patterns in VEXAS from healthy donors (Supplementary Fig. 2d).

### Dysfunctional monocytes in VEXAS
To explore innate immune cell states in VEXAS, 24,887 monocytes and 704 DCs were isolated (after excluding contaminated small cell clusters). Cells were subclustered into five subtypes, including classical monocytes, intermediate monocytes, non-classical monocytes, conventional DCs (cDC), and plasmacytoid DCs (pDC), based on transcriptional profiles of sorted bulk data sets and expression of canonical monocyte/DC gene markers. (Fig. 3a and Supplementary Fig. 3a)[15]. Differential abundance analysis showed cDCs, pDCs, and non-classical monocytes to be significantly reduced in VEXAS (Fig. 3b, c), consistent with previous reports[1]. VEXAS monocytes showed downregulated HLA class II genes and upregulated alarmin-related S100A genes (Fig. 3d; Supplementary Data 3). Gene set enrichment analysis (GSEA) of monocytes and cDCs from VEXAS (pDCs were removed due to a low number of cells in cases) showed upregulation of multiple inflammatory response and apoptosis pathways across all cell subtypes compared to those from healthy donors (Fig. 3e).

Although none of the patients had active bacterial infection at the time of blood collection, a gene expression profile of monocytes from VEXAS (increased alarmin-related S100A gene and decreased HLA class II gene expression) was similar to that of dysfunctional monocytes that have been described in severe sepsis[16]. Integration of our scRNA-seq data with published scRNA-seq data derived from monocytes of patients with multisystem autoimmune inflammatory diseases [including systemic lupus erythematosus (SLE), microscopic polyangiitis (MPA), and Behcet's disease (BD)][17–19] was performed to compare relative expression levels of HLA class II genes, sepsis-associated

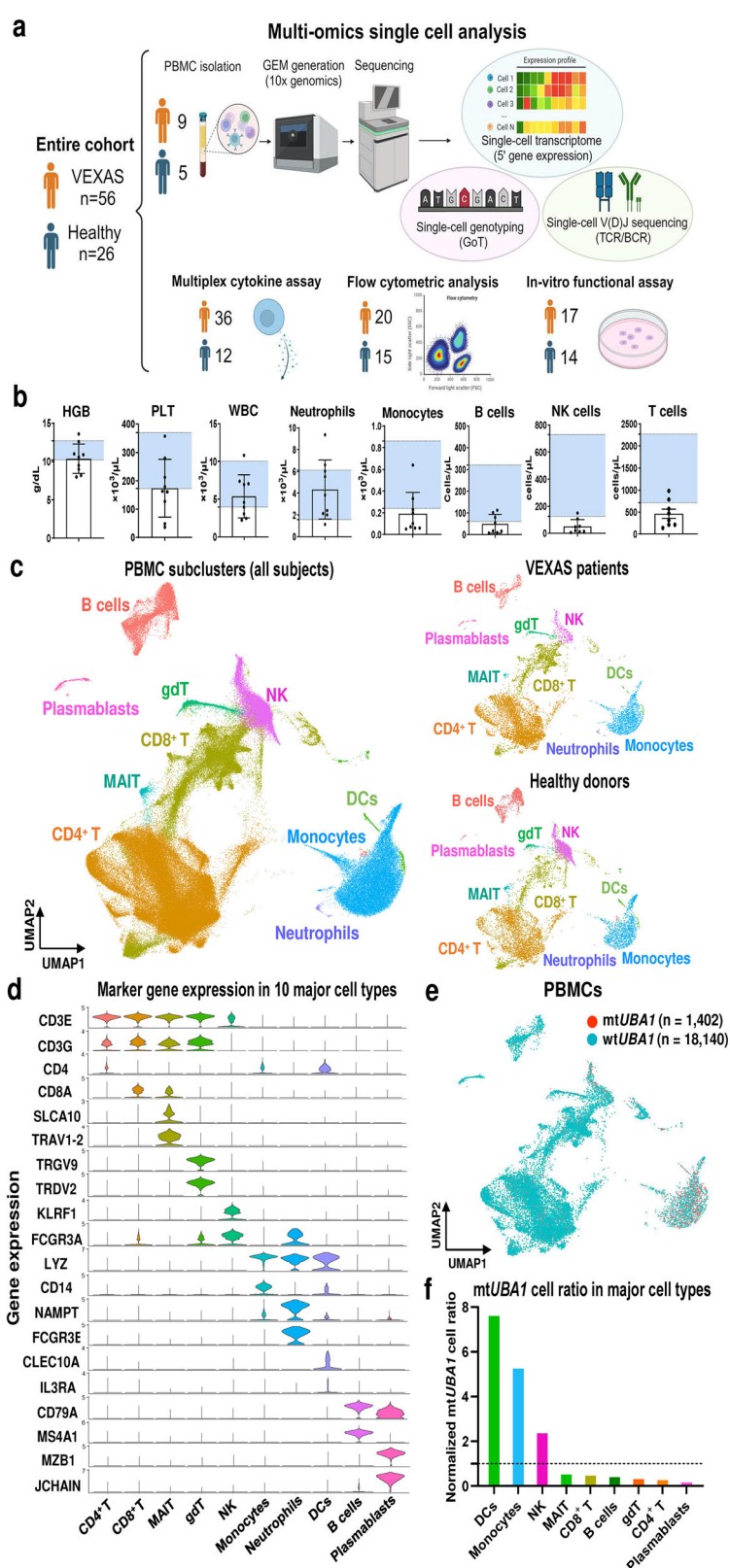

monocyte signature genes[16], and genes related to S100A alarmins and inflammatory responses across different autoimmune diseases. We observed pronounced decreases in HLA class II gene module scores, increases in sepsis-associated monocyte signature gene and alarmin-related S100A gene scores in VEXAS, when compared to those in healthy donors and in other autoimmune diseases (Fig. 3f and Supplementary Fig. 3b). In contrast, gene module scores for the IFN-α, IFN-

γ, and TNF response pathways were similar among VEXAS and other autoimmune diseases (Supplementary Fig. 3b).

To investigate whether monocytes dysregulation in VEXAS was specific to mtUBA1 cells, GoT analysis was employed to identify UBA1 transcripts, resulting in detection of UBA1 transcripts in 1632 out of 16,821 monocytes (651 mtUBA1 and 981 wtUBA1 monocytes) (Fig. 3g). mtUBA1 monocytes were evenly distributed in all monocyte subtypes,

**Fig. 1 | Multi-omics single-cell sequencing analysis of PBMCs from VEXAS patients and healthy donors. a** Overview of the experimental workflow. A figure was created with BioRender. Mizumaki, H. (2025) [https://BioRender.com/n37z398]. **b** Hemoglobin levels (HGB), platelet counts (PLT), white blood cell counts (WBC), neutrophil counts, monocyte counts, B cell counts, NK cell counts, and T cell counts from VEXAS patients (n = 9). Background shading shows a normal reference range for each parameter. Data are presented as mean and SD. **c** A Uniform Manifold Approximation and Projection (UMAP) plot of 178,168 cells from all subjects (n = 14, left). UMAP plots of 125,806 peripheral blood mononuclear cells (PBMCs) derived from VEXAS patients (n = 9, upper right) and 52,362 PBMCs derived from healthy donors (n = 5, lower right). Leiden clusters based on 5′ gene expression are shown and colored by major cell types. **d** A violin plot showing expression distributions of selected canonical marker genes in 10 major cell types. Rows and columns represent selected marker genes and cell types, respectively. **e** A UMAP plot of cells with projected mutation status assignment for wild-type *UBA1* (wt*UBA1*; n = 18,140 cells) and mutated *UBA1* (mt*UBA1*; n = 1402 cells). **f** Normalized frequency of mt*UBA1* cells in major cell types. Cell types with more than 100 cells were analyzed. gdT gamma delta T cells, MAIT mucosa-associated invariant T cell. Source data are provided as a Source Data file.

and a ratio of mt*UBA1* monocytes along pseudotime trajectories was similar to that of wt*UBA1* monocytes (Fig. 3h and Supplementary Fig. 3c). Gene module scores of the apoptosis pathway in wt*UBA1* monocytes tracked by pseudotime were similar to those of mt*UBA1* monocytes, and both were higher than for monocytes from healthy donors (Supplementary Fig. 3d). Unexpectedly, there were very few differentially expressed genes between mt*UBA1* and wt*UBA1* monocytes; no differences were observed in both the sepsis-associated monocyte signature gene and HLA class II gene module scores, suggesting a global dysfunctional gene signature of VEXAS monocytes, irrespective of the *UBA1* genotype (Supplementary Fig. 3e, f; Supplementary Data 3).

To functionally validate monocyte dysregulation, we measured efferocytosis, a process by which apoptotic cells are removed by "professional" phagocytic cells[20], using CD14+ monocytes isolated from an additional six VEXAS patients and six healthy donors (Supplementary Data 1). After induction of apoptosis with Staurosporine, CSFE-labeled apoptotic "bait" neutrophils from an unrelated healthy donor were fed to "effector" monocytes (Supplementary Fig. 4a, b). Monocytes from VEXAS patients exhibited a significant decrease in the frequency of CSFE+ apoptotic cell uptake (Fig. 3i and Supplementary Fig. 4b), indicating impaired efferocytosis. Additionally, cytokine responses of monocytes from VEXAS patients measured after lipopolysaccharide (LPS) stimulation showed significantly blunted cytokine release relative to those from healthy donors (Fig. 3j and Supplementary Fig. 4c). Blunted cytokine production by monocytes from VEXAS patients was also observed after stimulation with other toll-like receptor (TLR) ligands (Supplementary Fig. 4d), demonstrating that monocytes of VEXAS patients were functionally impaired.

### Inflammatory mt*UBA1* NK cells in VEXAS patients

Despite the absence of *UBA1* mutations in B and T cells, some VEXAS patients have small fractions of mt*UBA1* NK cells in PB[11]. To elucidate NK cell features in VEXAS, 11,846 NK cells were subclustered into four subsets, including CD56bright, early CD56dim, CD56dim, and adaptive-like NK cells, according to previously published transcriptional signatures of NK cells (Fig. 4a)[21]. The adaptive-like NK cells featured high expression of *CD3E*, *LAG3*, and *IL-32* with no expression of *FCER1G* and TCRs, consistent with published scRNA-seq data (except for low expression of *KLRC2/NKG2C*, the canonical markers for adaptive NK cells after human cytomegalovirus (HCMV) infection) (Supplementary Fig. 5a, b)[21,22]. Differential abundance analysis identified elevated adaptive-like NK cells in VEXAS (Fig. 4b, c); flow cytometry confirmed the scRNA-seq data (Supplementary Fig. 5c, d).

GoT analysis clarified NK cell features linked to *UBA1* mutations: *UBA1* transcripts were identified in 669 out of 3867 NK cells (125 mt*UBA1* and 574 wt*UBA1* NK cells) (Fig. 4d). In contrast to monocytes, we observed progressive loss of mt*UBA1* NK cells along pseudotime trajectories (Fig. 4e and Supplementary Fig. 5e). Upregulation of gene pathways associated with inflammation and apoptosis were seen across all NK subtypes in VEXAS (Fig. 4f), but differences of gene module scores of those gene pathways between patients and healthy donors decreased with differentiation (Supplementary Fig. 5f), indicating that upregulation of the inflammatory and apoptosis pathways

was dependent on a proportion of mt*UBA1* cells in each NK cell subtype. When gene expression levels were compared between mt*UBA1* and wt*UBA1* NK cells, genes associated with the interferon response pathways (*IFITM1*, *IFITM2*, and *AREG*) and the TNF response pathway (*TNFRSF4*, *TNFRSF18*, and *IL2RB*) were enriched in mt*UBA1* NK cells (Fig. 4g; Supplementary Data 4). In comparison of gene module scores between mt*UBA1* and wt*UBA1* NK cells, there was cell-intrinsic upregulation of the inflammatory and cytotoxic activity pathways with enhanced apoptosis in mt*UBA1* NK cells (Fig. 4h).

### Clonal expansion and activation of effector memory CD8+ T cells in VEXAS

To better characterize T cells in VEXAS, 128,236 T cells were subclustered according to transcriptional profiles of sorted bulk data sets and expression of canonical T cell gene markers[23]. Ten T cell subtypes were identified: naïve CD4+ T cells, central memory CD4+ T cells, effecter memory CD4+ T cells, naïve CD8+ T cells, central memory CD8+ T cells, effecter memory CD8+ T cells, MAIT, gdT, proliferative T cells (proT), and regulatory T cells (Treg) (Fig. 5a and Supplementary Fig. 6a). By differential abundance analysis, frequencies of central and effector memory CD8+ T cells, and Treg were elevated, and effector memory CD4+ T cells and MAIT were reduced (Fig. 5b, c).

scTCR-seq based on scRNA-seq libraries identified 119,850 cells with productive TCRs (Supplementary Fig. 1b). V(D)J sequences of TCR repertoires were analyzed to assess clonal relationships among individual T cells. In almost all T cell subsets, more than 60% of cells expressed TCRs, except for MAIT and gdT subsets (Fig. 5d). Large clonal expansions were observed particularly in effector CD8+ T cells and more pronounced in VEXAS patients than in healthy donors (Fig. 5d, e). We determined T cells clonal expansion index by both the Gini index (a measure of the "unevenness" of the number of RNA molecules per unique VDJ region sequence) and the Shannon diversity index (a measure of the unevenness of unique VDJ region sequences per clone). CD8+ T cells from VEXAS patients had higher Gini index and lower Shannon diversity index compared to those from healthy donors, indicating restricted TCR usage in CD8+ T cells in VEXAS (Fig. 5f). Examination of CDR3 homology, which might implicate a potential common antigen, showed little sharing of TCR clones among VEXAS patients, or with healthy donors (Supplementary Fig. 6b). When GLIPH2[24], an algorithm for clustering TCRs based on amino-acid level similarities, was used to identify common TCR groups and differential abundance in VEXAS, no TCR groups specific to VEXAS were found (Supplementary Data 5). Thus, TCR usage in VEXAS was private rather than disease-specific; no shared target antigens driving clonal expansions of CD8+ T cells in VEXAS were identified.

Immune response pathways, especially the IFN-γ response pathway, were upregulated across CD8+ T cell subtypes in VEXAS patients, all mostly wt*UBA1* (Supplementary Fig. 6c). Genes related to the IFN-γ response pathway, T-cell mediated cytotoxicity, and T cell exhaustion were upregulated along pseudotime trajectories, more prominent in VEXAS patients than in healthy donors (Fig. 5g and Supplementary Fig. 6d). Expanded CD8+ T cell clones, defined as >20 cells with identical CDR3 sequences, showed higher IFN-γ response, T-cell mediated cytotoxicity, and exhaustion scores than did expanded reactive CD8+ T cells from

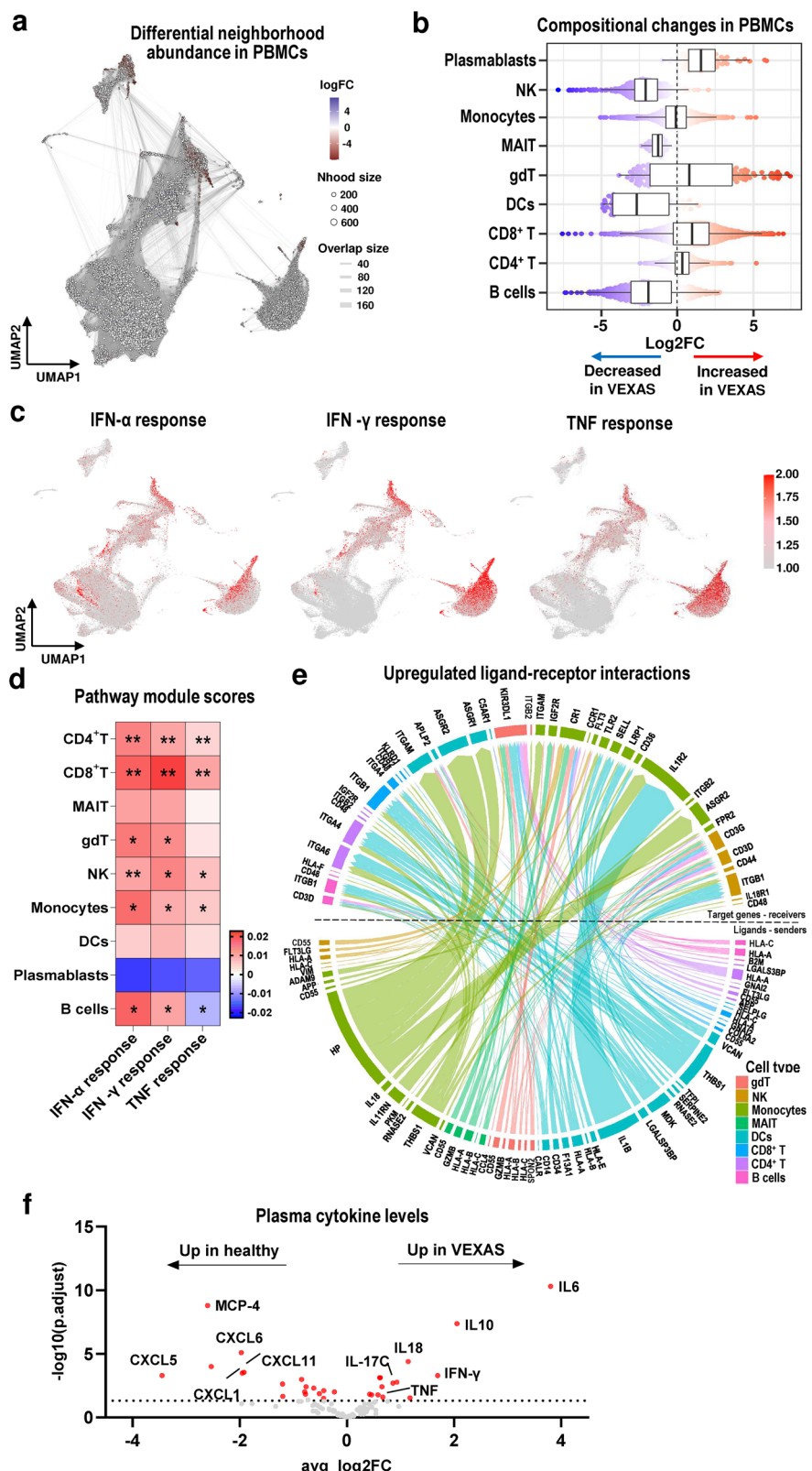

**a** Differential neighborhood abundance in PBMCs

**b** Compositional changes in PBMCs

**c** IFN-α response | IFN-γ response | TNF response

**d** Pathway module scores

**e** Upregulated ligand-receptor interactions

**f** Plasma cytokine levels

healthy donors (Supplementary Fig. 6e), indicating more pronounced adaptive inflammatory response.

### Clonal expansion and skewed differentiation of B-lineage cells in VEXAS

Plasma cell dyscrasias, monoclonal B cell lymphocytosis, and multiple myeloma are common in VEXAS patients, even though VEXAS patients commonly exhibit B cell lymphocytopenia and their B cells rarely harbor the *UBA1* mutations[1,11]. To further define B cell features in VEXAS, 11,335 B cells were subclustered into six subsets including naïve, IgM memory, classical memory, double negative, and plasmablasts, according to previously reported transcriptional profiles of sorted bulk data sets and the expression of canonical gene markers (Fig. 6a and Supplementary Fig. 7a)[25]. Differential abundance analysis

**Fig. 2 | Compositional changes and activation of the inflammatory pathways in immune cells from VEXAS patients. a** A neighborhood graph of PBMCs using Milo differential abundance testing. Nodes represent neighborhoods from the PBMC population. A color scale indicates log2-fold change (FC) differences between VEXAS patients and healthy donors. Significant changes are colored in blue and red. Nondifferential abundance Nhoods (a false discovery rate [FDR] ≥ 0.10) are indicated in white. **b** Beeswarm and box plots showing the distribution of log2FC differences between VEXAS (n = 9) and healthy donors (n = 5) in neighborhoods in different cell type clusters. Colors are represented similarly to (**a**). A box plot shows median and interquartile ranges (IQR); lower and upper hinges correspond to the first and third quartiles, respectively. An upper whisker extends from the hinge to the largest value no further than 1.5*IQR from the hinge. A lower whisker extends from the hinge to the smallest value at most 1.5*IQR from the hinge. **c** UMAP plots overlaid with projections of the IFN-α, IFN-γ, and TNF response pathway gene

module scores. **d** Gene module scores of IFN-α, IFN-γ, and TNF responses in PBMCs. A heatmap depicting differences between average scores of VEXAS patients and those of healthy donors in each of 10 cell types. Gene module scores of cells of VEXAS patients were compared to those of healthy donors in each cell type using the two-sided Welch's $t$-test and shown as $*P < 1.0 \times 10^{-10}$, $**P < 1.0 \times 10^{-100}$. **e** Predicted ligand-receptor interactions of significantly upregulated genes in VEXAS patients as compared to healthy donors. **f** A volcano plot displaying normalized protein expression of cytokines in plasma measured by the Olink Target 96 Inflammation panel immunoassay. The plot shows 92 plasma cytokines differentially expressed between VEXAS patients and healthy donors. x and y axes represent a magnitude of a cytokine's log2FC and a significance scale by the -log10 ($p$ adjusted [$p$adj]), respectively. A black dotted line indicates $p$adj = 0.05. Significantly dysregulated cytokines are highlighted in red. gdT gamma delta T cells, MAIT mucosa-associated invariant T cell. Source data are provided as a Source Data file.

---

identified prominent decreases in transitional B cells and naïve B cells, and an increase in plasmablasts in VEXAS patients (Fig. 6b, c); flow cytometry confirmed the scRNA-seq data (Fig. 6d and Supplementary Fig. 7b, c). scBCR-seq identified 9799 cells with productive BCRs (Supplementary Fig. 1b). Increased numbers of expanded BCR clonotypes were found in VEXAS patients, especially in plasmablasts, relative to healthy donors (Fig. 6e, f). When V(D)J gene usage was compared between VEXAS and healthy donors, there was no obvious bias in IGHV gene usage in B cells in disease (Supplementary Fig. 7d).

scBCR-seq data were analyzed for Ig repertoires. Various immunoglobulin (Ig) isotypes distributions (IgA, IgD, IgG, and IgM, but not IgE) and their somatic hypermutations (SHM) were evaluated in each B cell subtype; transitional B cell populations were excluded from analysis due to the extremely small numbers in patients. Consistent with the canonical gene profile, Ig sequences expressed by naïve B cells were IgD or IgM in isotypes with low SHM (mutation frequency ≤ 3%) or no SHM (Fig. 6g and Supplementary Fig. 7e, f). Memory B cells expressed Igs of a diverse array of isotypes, including IgA1, IgA2, IgG1, IgG2, IgG3, IgM, and to a lesser extent IgD with high SHM (mutation frequency > 3%).

While differences in isotype distributions of both IgM and classical memory B cells were modest between VEXAS patients and healthy donors, a higher level of SHM was observed in IgM memory B cells from VEXAS patients in comparison to healthy donors (Fig. 6g and Supplementary Fig. 7f, g). A small proportion of transcriptionally-defined IgM memory B cells expressed non-IgM Igs, and differences in SHM were still evident even analyzing only IgM-expressed memory B cells (Supplementary Fig. 7h). In contrast to the modest differences in isotype distributions observed in naïve and memory B cells, plasmablasts in VEXAS exhibited significant IgG bias with highSHM compared to those in healthy donors (Fig. 6g and Supplementary Fig. 7g).

Transcriptional features of plasmablasts from VEXAS patients, largely wt*UBA1*, were further assessed, identifying significant upregulation of genes involved in protein processing in ER, the ER stress response pathways (such as MTOR1 signaling and the UPR pathway), and the cell-cycling pathways in VEXAS in comparison to healthy donors (Fig. 6h). These results indicate enhanced antibody production and cell-proliferation in plasmablasts from VEXAS patients.

In summary, VEXAS B cells were characterized by skewed differentiation from memory B cells toward a mature phenotype with loss of transitional B cells and increased plasmablasts displaying a significant IgG bias, high SHM, and upregulation of protein processing genes.

## Discussion

VEXAS is a clonal myeloid disease with heterogeneous systemic manifestations that overlap many established hematological and rheumatologic diseases. While inflammation remains the main presentation in those with canonical *UBA1* mutations, there is variability in the organs involved, glucocorticoid requirements, response to biologic therapies, as well as overall disease course among patients. Besides genotype-

phenotype associations based on residual cytoplasmic UBA1b isoform[11], there is little explanation for variable penetrance and target organ involvement in VEXAS. Mutations in *UBA1* disturb critical pathways in cellular homeostasis. Indeed, *UBA1* mutations are tolerated differently by different cellular compartments in the marrow and have distinct effects on functionality[26,27]. In this study, we aimed to assess contributions of mature innate and adaptive immune cells to disease in VEXAS using multi-modal single-cell techniques.

VEXAS patients presented a broad immune activation with upregulation of inflammatory response pathways in all PB cell types, irrespective of the *UBA1* genotype, as well as increased levels of pro-inflammatory cytokines. Both wt*UBA1* and mt*UBA1* monocytes in VEXAS exhibited dysfunctional transcriptional features. Functionally, these monocytes showed impaired efferocytosis and blunted cytokine production. Despite progressive loss in PB, mt*UBA1* NK cells also showed upregulation of genes associated with inflammatory response and cytotoxicity pathways. T and B cell compartments were characterized by clonal expansion of CD8[+] effector memory T cells with high cytotoxicity and exhaustion levels, and increased clonal plasmablasts with enhanced antibody production and cell proliferation, respectively.

In VEXAS, mature myeloid cells, including neutrophils, monocytes, and DC cells, are largely mt*UBA1* and expected to be the main drivers of systemic inflammation. Although both neutrophils and monocytes harbor *UBA1* mutations, in the marrow, there is a differentiation bias towards neutrophils over monocytes with increased expression in the *CEBPA* master transcription factor and decreased expression in *IRF8*[10]; with notable monocytopenia in 30% of patients[7]. Consistent with a previous report[26], in PB, DCs were severely decreased, and monocytes, despite of a pro-inflammatory transcriptional profile with upregulation of multiple immune response pathways, were dysfunctional. Monocytes from VEXAS patients had a distinct but overlapping transcriptional signature with severe sepsis and other autoimmune diseases; HLA class II genes were downregulated while alarmin-related S100A genes were upregulated. Most of the patients in the scRNA-seq cohort were in the late phase of their disease (median time from symptom onset was 7.1 [range, from 1.9 to 10.1] years), and prolonged inflammation itself may have diminished the capacity of VEXAS monocytes to release proinflammatory cytokines in response to additional exposure to stimuli, despite intrinsically upregulated immune response pathways. Despite dysfunctional features, monocytes were highly interactive with other immune cells, as inferred computationally from ligand-receptor interactions.

Efferocytosis was blunted in monocytes from patients, irrespective of *UBA1* mutation status. Efferocytosis is critical for tissue homeostasis, and defective efferocytosis results in the accumulation of apoptotic cells, secondary necrosis, and release of pro-inflammatory cellular contents such as damage-associated molecular patterns (DAMP)[20]. Inadequate efferocytosis may be another pathogenic contributor to inflammation in VEXAS. Quantitative and qualitative defects

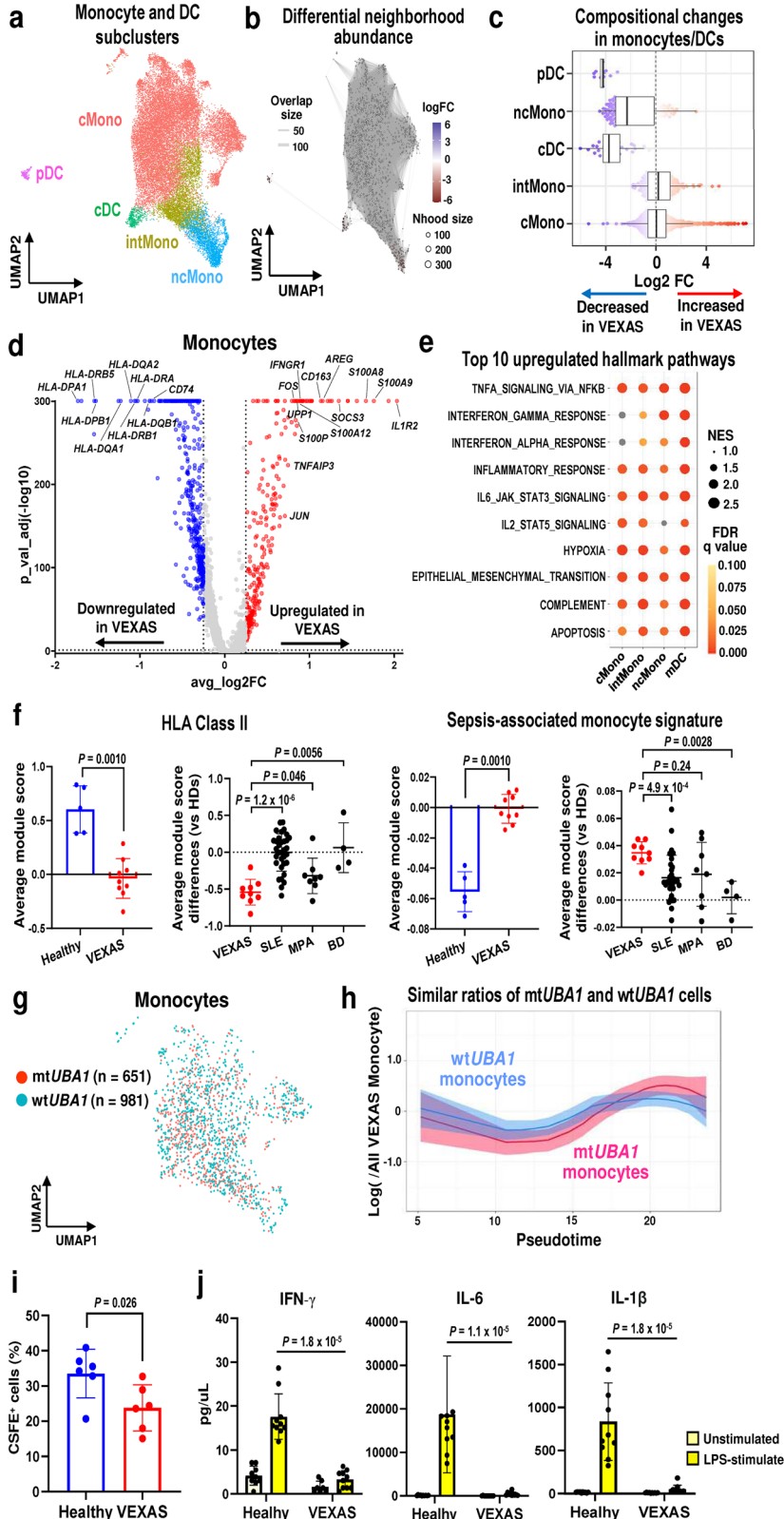

in monocytes may also explain the increased risk of opportunistic serious infections by intracellular pathogens, such as atypical mycobacterial infections and legionellosis, in VEXAS[28–31].

In contrast to monocytes, *UBA1* mutations directly impacted the transcriptional profile of a subset of NK cells. NK cells originate from lymphoid progenitors, but they participate in cytolytic innate and antigen-specific adaptive immune responses[32]. Similar to mt*UBA1*

lymphoid progenitor cells[10], mt*UBA1* NK cells progressively decreased with differentiation compared to wt*UBA1* NK cells. Comparison of gene expression between mt*UBA1* and wt*UBA1* NK cells uncovered elevated expression of the inflammatory and cytotoxic pathways in mt*UBA1* NK cells. CD56[bright] NK cells, efficient cytokine producers, were mostly mt*UBA1* and could also contribute to auto-inflammation in VEXAS. In contrast, adaptive-like NK cells, mostly wt*UBA1*, were expanded in

**Fig. 3 | Dysfunctional monocytes in VEXAS. a** A UMAP plot of 24,887 monocytes and 704 dendritic cells (DC). **b** A neighborhood graph of monocytes/DCs, which was generated similarly as in Fig. 2a. **c** Beeswarm and box plots of monocytes/DCs for VEXAS (n = 9) and healthy donors (n = 5), which were generated similarly as in Fig. 2b. **d** A volcano plot of differentially expressed genes of monocytes in VEXAS as compared to healthy donor: upregulated and downregulated genes in red and blue, respectively. A horizontal dotted line represents $p$adj = 0.05, vertical dotted lines indicate absolute log2FC values = +−0.25. $P$ values were calculated with the two-sided Wilcoxon rank-sum test and Bonferroni correction for multiple comparisons. **e** A dot plot showing gene set enrichment scores of top 10 upregulated hallmark pathways across subtypes. Dot sizes indicate mean normalized enrichment score (NES) differences between VEXAS and healthy donors, and the color scale indicates FDR values; non-significant pathways (FDR ≥ 0.10) in gray. **f** Gene module scores of HLA class II genes and sepsis-associated monocyte signatures in monocytes across VEXAS (n = 9), healthy donors (n = 5), and other multisystem autoimmune diseases (Systemic lupus erythematosus (SLE), n = 33; microscopic polyangiitis (MPA), n = 8;

Behcet's disease (BD), n = 4). Data are presented as mean with SD. $P$ values were calculated with the two-sided unpaired Mann–Whitney $U$ test. **g** A UMAP plot of cells with projected $UBA1$ mutation status as wt$UBA1$ (981 cells) and mt$UBA1$ monocytes (651 cells). **h** Dynamic changes of mt$UBA1$ and wt$UBA1$ monocytes ratios to all monocytes in VEXAS patients (log scale on y axis) along differentiation (pseudotime ordering from classical to non-classical monocytes on x axis). Data are presented as mean with 1.96*SE. **i** Frequency of CSFE⁺ bait cells among CytoTell Blue⁺CD14⁺ monocytes (VEXAS, n = 6; healthy donors, n = 6). Data are presented as mean with SD. $P$ values were calculated using the two-sided unpaired Mann–Whitney $U$ test. **j** Cytokine detection of IFN-γ, IL-6, and IL-1β in culture supernatants of purified CD14⁺ monocytes (VEXAS, n = 10; healthy donors, n = 10). Data are presented as mean with SD. $P$ values were calculated using the two-sided unpaired Mann–Whitney $U$ test. cMono classical monocytes, intMono intermediate monocytes, ncMono non-classical monocytes, cDC conventional DCs, pDC plasmacytoid DCs. Source data are provided as a Source Data file.

VEXAS compared to healthy donors, but they exhibited a non-inflammatory transcriptomic profile.

Clonal expansions of specific T and B cell subsets were observed in VEXAS. As previously shown in BM[10], CD8⁺ T cells in PB of VEXAS were clonally expanded with upregulated IFN-γ and cytotoxicity pathways. Within the B cell compartment, there was a striking increase in clonal plasmablasts even in patients with severe B cell lymphopenia; expanded clones displayed high SHM. Autoantibodies produced by these SHM cells have been associated with development of some autoimmune diseases[33,34]. An increased ratio of plasmablasts to total B cells in PB occurs in ANCA-associated vasculitis (AAV) and SLE, accompanied by elevated levels of autoantibodies such as anti-dsDNA[35,36]. Cases of VEXAS associated with SLE and AAV have been reported, and the presence of lupus anticoagulant is not uncommon[8,37–39]. Screening of autoantibodies in large VEXAS cohorts, in comparison with other autoimmune diseases, would help understand a role of autoreactive B cells in the syndrome.

Without evidence of disease-specific antigens, nonspecific antigens produced by mt$UBA1$ myeloid cells as a result of dysregulated protein degradation may trigger clonal expansion of lymphoid cells in VEXAS. Since plasmablasts are short-lived and antibody-secreting B cells mature into plasma cells, clonal expansion of these cells may also account for monoclonal gammopathy that is found in 1/3 of VEXAS patients.

Another feature of the B cell compartment in VEXAS was the depletion of transitional B cells, which are thought to represent a link between immature and mature B cells in the marrow and the periphery[40]. As $UBA1$ mutations are absent in mature B cells but present in most lymphoid progenitor cells in BM[11], mt$UBA1$ B cells appear to be negatively selected during transitional B cell maturation in BM, producing a deficit of terminal cells. Transitional B cells, phenotypically and functionally related to IL-10-producing regulatory B cells, play regulatory roles under inflammatory conditions, and their loss may also contribute to the uncontrolled autoinflammation in VEXAS[40].

Immune dysregulation irrespective of $UBA1$ mutations was observed in lymphocytic cells that are often decreased in PB of VEXAS patients; we observed upregulation of the apoptosis and TNF response pathways in PB cells. As reported for the cytotoxicity of upregulated UPR pathways[27], effects of TNF are highly cell-type dependent; HSCs are resistant to TNF cytotoxicity, but TNF induces apoptosis in committed progenitor and mature cells[41]. In VEXAS, these immune compartments might be subjected to apoptosis due to chronic overactivation of the TNF response pathway. In contrast, neutrophils generally remain normal to elevated throughout the disease course and are predominant in biopsies of affected tissues[11,42]. We found decreased levels of many chemokines in the blood of VEXAS patients, especially CXCL5, which is generated by lung and other epithelial cells under inflammation and acts as a potent chemoattractant and

an activator of neutrophils via CXCR1 and CXCR2 receptors[43,44]. Dysregulated chemokine expression in blood, with the upregulation of ligand-receptor interactions involved in cell adhesion in VEXAS, indicates migration of immune cells from blood to tissues, explaining the infiltration of mature myeloid cells in inflamed tissues such as the lung and skin.

Broad immune activation with upregulation of multiple immune response pathways was observed across all immune cells, especially NK cells, CD8⁺ T cells, and monocytes, with increased levels of multiple inflammatory cytokines, such as IL-6, IFN-γ, TNF, and IL-18 in plasma. The immunological networks by connectome analysis depicted abundant cellular interactions among innate and adaptive immune cells. These data cross-validate findings from other approaches and highlight the scope of VEXAS hyper-immunity; they are relevant for therapeutic approaches in VEXAS, and explain failures of targeted agents such as TNF and IL-1 blockade[45]. Non-targeted treatments such as hypomethylating agents and JAK inhibitors produce better clinical response[46–49].

Our study has some limitations. First, all VEXAS patients were treated with corticosteroids, with some other agents to control severe autoinflammation; such therapies could affect responses and cytotoxicity in immune cell samples. Nevertheless, VEXAS patients showed upregulation of multiple inflammatory pathways as compared to healthy donors. In addition, GoT analysis allowed us to compare mt$UBA1$ and wt$UBA1$ cells, which were equally subjected to anti-inflammatory drugs. Second, neutrophils could not be investigated due to the low number of cells captured in our scRNA-seq experiments. scRNA-seq analysis of neutrophils is needed for better understanding of the VEXAS pathophysiology. Third, female VEXAS patients are extremely rare, limited to individuals who have monosomy X[50,51], there is obvious sex bias.

In summary, systemic inflammation in VEXAS is driven by innate and adaptive immune dysregulation irrespective of $UBA1$ mutations. Our scRNA-seq data, validated by an independent cohort using traditional immunological methods such as flow cytometry and in vitro functional assays, support the hypothesis that autoinflammation initiated by mt$UBA1$ myeloid cells significantly impacts residual wt$UBA1$ lymphoid cells, perpetuating a vicious cycle of harmful auto-immunity. Our work provides a deeper understanding of VEXAS pathophysiology and can serve as a guide for therapeutic development in this complex disease.

## Methods

### Human samples

PB samples were obtained from VEXAS patients after written informed consent under the protocol (www.clinicaltrials.gov NCT05012111) approved by the Institutional Review Boards of the National Heart, Lung, and Blood Institute (NHLBI), in accordance with the Declaration of

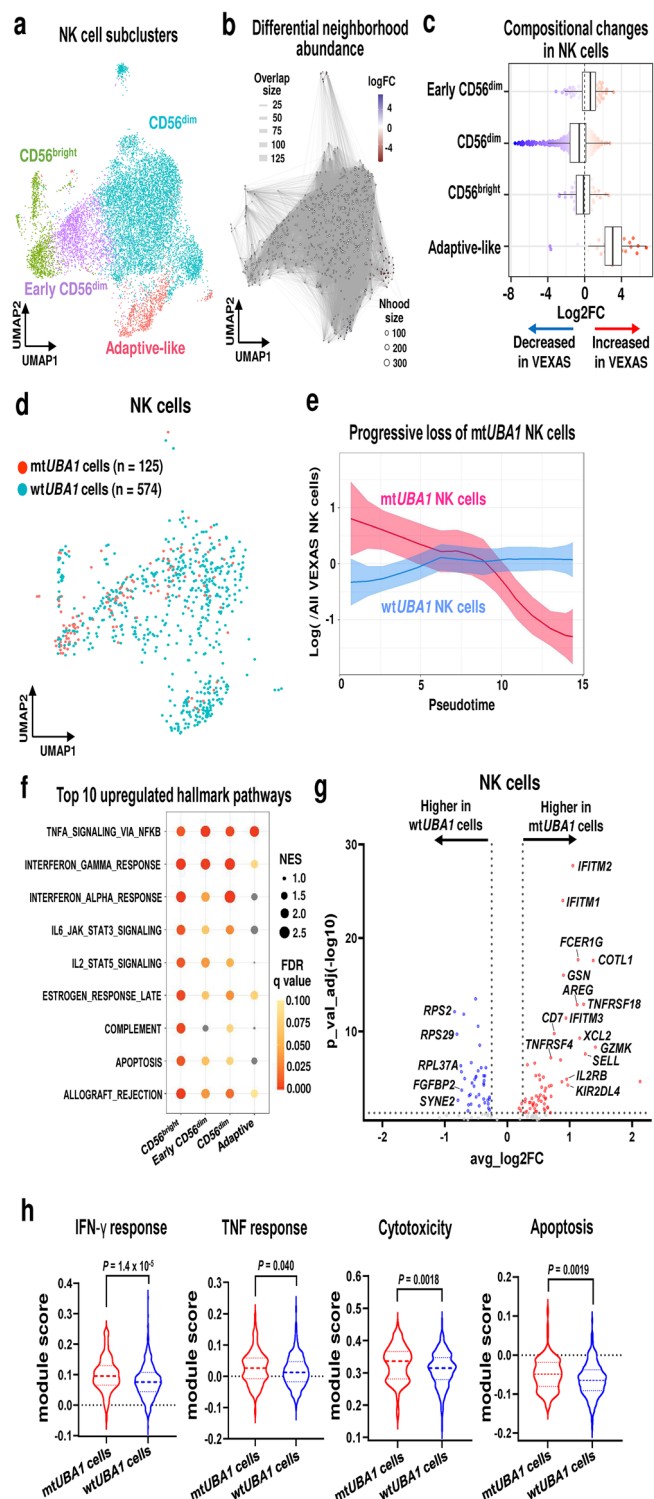

**Fig. 4 | Compositional and transcriptomic landscapes of mt*UBA1* NK cells.**
**a** A UMAP plot of 11,846 NK cells. **b** A neighborhood (Nhood) graph of NK cells, which was generated similarly as in Fig. 2a. **c** Beeswarm and box plots of NK cells for VEXAS (n = 9) and healthy donors (n = 5), which were generated similarly as in Fig. 2b. **d** A UMAP plot of cells with projected *UBA1* mutation status assignment for wt*UBA1* (574 cells) and mt*UBA1* NK cells (125 cells). **e** Dynamic changes of mt*UBA1* and wt*UBA1* NK cell ratios to all NK cells in VEXAS patients along differentiation. x axis, pseudotime ordering from CD56^bright NK cells to adaptive-like NK cells estimated by slingshot; y axis, ratios of cell numbers of mt*UBA1* or wt*UBA1* NK cells to all VEXAS NK cells on a log scale. Data are presented as mean with 1.96*SE. **f** A dot plot showing gene set enrichment scores of top 10 upregulated hallmark pathways across subtypes. Dot sizes indicate mean normalized enrichment score (NES) differences between VEXAS patients and healthy donors, and the color scale indicates FDR values. Non-significant pathways (FDR ≥ 0.10) are in gray. **g** A volcano plot of differentially expressed genes between mt*UBA1* NK cells and wt*UBA1* NK cells. Genes upregulated and downregulated in VEXAS patients are highlighted in red and blue, respectively. A horizontal dotted line, a *p*adj value = 0.05; vertical dotted lines, absolute log2FC values = −0.25 and 0.25. **h** Gene module scores of the IFN-γ response, TNF response, leukocyte mediated cytotoxicity, and apoptosis pathways in mt*UBA1* and wt*UBA1*NK cells. *P* values were calculated with the two-sided unpaired Mann–Whitney *U* test. Source data are provided as a Source Data file.

mononuclear cell separation medium (#17544202, Cytiva). Briefly, PB samples diluted twofold with phosphate buffered saline (PBS) (#10010031, Thermo Fisher Scientific) were layered on top of 1 volume Ficall-Paque medium in a 50-ml Falcon tube and centrifuged at $1140 \times g$ for 20 min at room temperature with brake off. Isolated PBMCs were treated with ACK lysing buffer (#118-156-101, Quality Biological) for lysis of red blood cells, washed with PBS, and resuspended in IMDM (#12440-053, Thermo Fisher Scientific) + 2% fetal bovine serum (FBS, #12306 C, Sigma-Aldrich). Resuspended PBMCs were subjected to scRNA-seq. Aliquots of PBMCs were cryopreserved in 10% DMSO (#D2650-100ML, Thermo Fisher Scientific) in heat-inactivated FBS. Plasma was isolated by centrifugation of a EDTA-treated tube and cryopreserved in a −80 °C freezer for subsequent measurement of cytokines.

**Plasma cytokine processing and data analysis**
A multiplexed Olink Target 96 Inflammation (Olink Biosciences) panel including 92 proteins from 59 plasma samples of 36 VEXAS patients and 12 healthy donors were profiled. Values used in the analyses were normalized protein expression units (NPX). Differentially expressed cytokines were identified with the Mann–Whitney *U* test. PCA plots were generated using Olink Analyze (v3.7.0) (Olink Biosciences).

**Efferocytosis assay**
Frequency of CD14⁺ efferocytes was assessed by the Efferocytosis Assay kit (#601770, Cayman Chemical) and as previously described[52]. CD14⁺ monocytes were used as effectors, and apoptotic neutrophils as target cells. Ex-vivo CD14⁺ monocytes rather than differentiated macrophages were used in the assay due to low cell availability. For effector cells, CD14⁺ monocytes were isolated from cryopreserved PBMCs (1 × 10⁷ cells) by using CD14 MicroBeads (#130-118-906, Miltenyi Biotec) following manufacturer instructions. Briefly, 1 × 10⁷ PBMCs were thawed and incubated with 20 µl of microbeads and 80 µl of buffer at 4 °C for 15 min. After incubation, cells were washed with 3 ml of buffer and resuspended in 500 µl of buffer. Positive selection was performed using the MACS Separator (Miltenyi Biotec). After separation, cells were counted and stained with CytoTell™ Blue provided in the kit according to manufacturer instructions. In brief, cells were resuspended in 200 µl of buffer containing 1X CytoTell™ Blue (stock diluted 1:400 in buffer), incubated at 37 °C for 30 min, and washed three times with R10 (RPMI medium with 10% FBS and Penicillin-Streptomycin; Thermo Fisher Scientific), resuspend in R10 at 1 × 10⁶ cells/µl, and used for the efferocytosis assay.

Helsinki. Age- and sex-matched healthy donors were enrolled as controls under protocol NCT00442195 in NHLBI.

**Blood sample processing**
PB samples from patients and healthy donors were collected into ethylenediaminetetraacetic acid (EDTA) tubes. PB samples were processed within 4 h after collection, followed by construction of scRNA-seq libraries for VEXAS 9–17, or cryopreserved until use for the rest of VEXAS patients.

PB mononuclear cells (PBMC) were isolated by Ficoll-Hypaque density gradient centrifugation using Ficall-Paque Premium

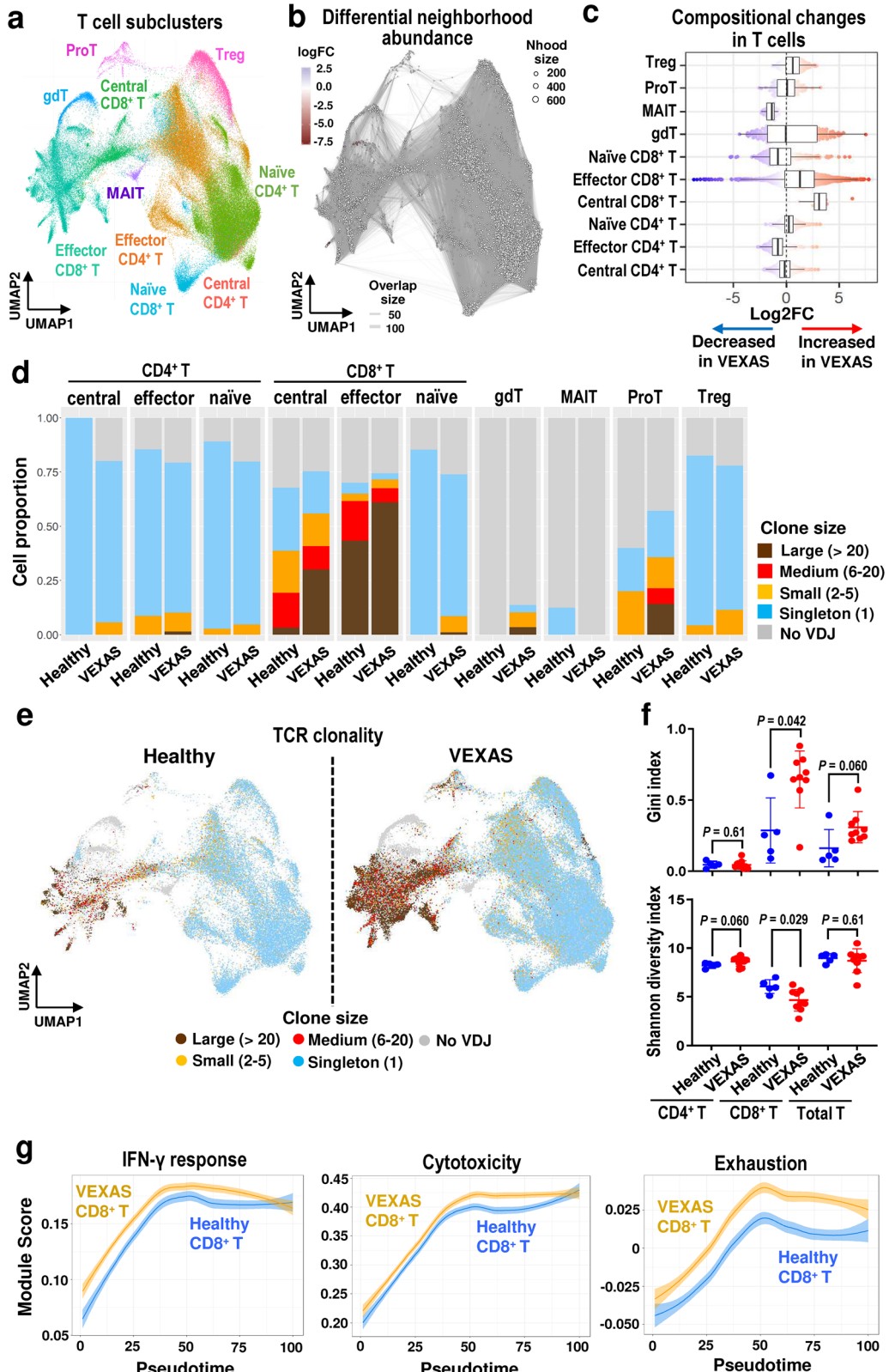

As target cells, neutrophils were isolated from blood of one unrelated healthy donor. Briefly, following isolation of PBMCs by Ficoll Plaque (#17544202, Cytiva), an equal volume of a solution of 20% dextran in water was added to a cellular pellet, gently mixed, and incubated for 1 min. After incubation, approximately three volumes of PBS were added, mixed again, and then incubated in the dark for 50–60 min. A clear top layer of the tube containing neutrophils was

collected, washed with PBS, and centrifuged at $1140 \times g$ for 10 min. Cells were treated with ACK lysing buffer at 37 °C for 5 min, washed with R10, and counted. Neutrophils were stained with CFSE provided by the kit following manufacturer instructions. Briefly, neutrophils were resuspended in buffer ($1 \times 10^7$ cells/ml), an equal volume of buffer containing 2X CFSE (a stock diluted 1:200 in buffer) was added to cells, incubated at 37 °C for 30 min, and washed three times with R10.

**Fig. 5 | Clonal expansion of effector CD8+ T cells with enhanced inflammation and cytotoxicity in VEXAS. a** A UMAP plot of 128,236 T cells. **b** A neighborhood (Nhood) graph of T cells, which was generated similarly as in Fig. 2a. **c** Beeswarm and box plots of T cells for VEXAS (n = 9) and healthy donors (n = 5), which were generated similarly as in Fig. 2b. **d** Distributions of T cell clone states in each subtype between VEXAS patients and healthy donors. **e** UMAP embedding of T cells from VEXAS patients (right) and healthy donors (left) colored by clonal expansion sizes. **f** Gini index and Shannon diversity index of TCR clonality in CD4+ T cells, CD8+ T cells, and total T cells from VEXAS (n = 9) and healthy donors (n = 5). Data are

presented as mean with SD. *P* values were calculated with the two-sided unpaired Mann−Whitney *U* test. **g** Dynamic changes of gene module scores of the IFN-γ response, T-cell mediated cytotoxicity, and exhaustion pathways in CD8+ T cells from VEXAS patients and healthy donors along differentiation. x axis, pseudotime ordering from naïve CD8+ T cells to effector memory CD8+ T cells estimated by slingshot; y axis, gene module scores for each pathway. Data are presented as mean with 1.96*SE. gdT gamma delta T cells, MAIT mucosa-associated invariant T cell, proT proliferative T, Treg regulatory T cells, TCR T cell receptor. Source data are provided as a Source Data file.

Apoptosis of neutrophils was induced by treatment with Staurosporine apoptosis inducer provided in the kit. Briefly, isolated cells were resuspended in R10 containing Staurosporine (a stock diluted 1:1000) and incubated at 37 °C for 3 h. After incubation, cells were washed two times with R10, resuspend in R10 at $1 \times 10^6$ cells/µl, and used for the efferocytosis assay. Effector CD14+ monocytes and target apoptotic neutrophils were cultured alone (as controls) or cocultured at a ratio of one effector CD14+ cells to three target apoptotic neutrophils. After incubation at 37 °C overnight (15 h), cells were washed with PBS, fixed with 1% paraformaldehyde in PBS, and acquired with a flow cytometer. Flow cytometry acquisitions were performed on the BD LSRFortessa (BD Biosciences) and examined using FACSDiva software (BD Biosciences) by acquiring all stained cells. Data was further analyzed using FlowJo v10.9 (TreeStar). Frequency of CD14+ efferocytes was determined as frequency of CFSE+ cells (neutrophils) in the CytoTellTM Blue+ cells (CD14+ cells), therefore representing frequency of CD14+ cells that engulfed apoptotic neutrophils. Supplementary Fig. 4b shows gating strategy: FSC/SSC/Sigle cells/CytoTell Blue+/CFSE+.

### In vitro stimulation of monocytes
Monocytes were isolated from frozen PBMCs by positive selection using the CD14 Microbeads (#130-118-906, Miltenyi Biotec) and plated at 25,000-30,000 cells/well in 96-well round-bottom plates. Monocytes were resuspended in RPMI medium (#11875093, Thermo Fisher Scientific) supplemented with 10% FBS (#12306 C, Sigma-Aldrich) and 1% Penicillin-Streptomycin (#15140148, Thermo Fisher Scientific), and stimulated in the absence or presence of LPS (100 ng/ml; #15140148, Innaxon) for 8 h at 37 °C. After stimulation, supernatants were harvested, centrifuged at 800 × *g* for 5 min at room temperature, and stored at −80 °C until use. Cytokine levels were measured by the multiplex bead-based immunoassay using the Immune Monitoring 65-Plex Human ProcartaPlex Panel (#EPX650-10065-901, R&D Systems), according to the manufacturer's instructions. To confirm the blunted cytokine production from VEXAS monocytes, purified CD14+ monocytes from additional VEXAS patients were stimulated in the absence or presence of LPS (100 ng/ml; #15140148, Innaxon), Pam3CSK4 (a synthetic ligand for TLR 1/2) (100 ng/ml, #tlrl-pms, InvivoGen), and R848 (a synthetic ligand for TLR7/8) (2.5 µM, #tlrl-r848-1, InvivoGen) for 8 h at 37 °C. After stimulation, supernatants were harvested, centrifuged at 800 × *g* for 5 min at room temperature, and stored at −80 °C until use. Cytokine levels were measured by the multiplex bead-based immunoassay using the Human Luminex® Discovery Assay (R&D Systems), according to the manufacturer's instructions. Data were acquired on the LUMINEX-200 (Thermo Fisher Scientific) instrument and analyzed by Bio-Plex Manager 6.1.1 software (Bio-Rad).

### Flow cytometry
To identify NK cells, cryopreserved PBMCs (5–10 × 10⁶ cells) were thawed and stained with the following fluorochrome-conjugated monoclonal antibodies (mAbs) in a total volume of 100 µl: APC-Cy7 anti-CD3 (clone SK7; #344818, 2.5 µl), APC-Cy7 anti-CD14 (clone M5E2; #301820, 2.5 µl), PE-Cy7 anti-CD56 (clone 5.1H11; #362510, 2.5 µl), APC anti-CD16 (clone B73.1; #360705, 2.5 µl), BV510 anti-CD57 (clone QA17A04; #393313, 2.5 µl) from BioLegend; APC-Cy7 anti-

CD19 (clone SJ25C1; #557791, 5 µl) from BD Bioscience; and 7-AAD (#00-6993-50, 3 µl) from Thermo Fisher Scientific for dead cell exclusion. For intracellular staining of FcεR1γ, cells were fixed and permeabilized with the Cyto-Fast Fix/Perm Buffer Set (BioLegend) after surface markers' staining, subsequently stained with FITC anti-FcεRI Antibody, a γ subunit (#FCABS400F, 2 µl) from Millipore Sigma. NK cell populations were identified as CD56+Lin-(CD3&CD14&CD19). CD56bright NK cells were identified as CD16-CD56bright and CD56dim NK cells as CD16+CD56dim. Adaptive-like NK cells were identified as lowSSC/Singlets/Live/CD56+CD3-CD14-CD19-/CD16+CD56dim/CD57+FcεR1γ-, as published previously[53,54].

For B cells, cryopreserved PBMCs (5–10 × 10⁶ cells) were thawed and stained with fluorochrome-conjugated mAbs in a total volume of 100 µl: APC-Cy7 anti-CD3 (clone SK7; #344818, 2.5 µl), APC-Cy7 anti-CD14 (clone M5E2; #301820, 2.5 µl), FITC anti-CD20 (clone 2H7; #302303, 2.5 µl), PE anti-IgD (clone IA6-2; #348203, 2.5 µl), BV785 anti-CD27 (clone O323; #302832, 2.5 µl), BV605 anti-CD24 (clone ML5; #311123, 5 µl), and BV421 anti-CD19 (clone H1B19; #302233, 2.5 µl) from BioLegend; APC anti-CD38 (clone HB7; #340439, 2.5 µl) from BD Bioscience; and 7-AAD (#00-6993-50, 3 µl) from Thermo Fisher Scientific used to exclude dead cells. B cell populations were identified as CD19+Lin-(CD3&CD14&CD56). Mature B cells were identified as CD24+CD38-, transitional B cells as CD24+CD38+, and plasmablasts as CD24-CD38+. Mature B cells were further divided into four subtypes based on their CD24 and IgD expression: class-switched memory B (CD27+IgD-), class-unswitched memory B (CD27+IgD+), naïve B (CD27-IgD+), and double negative B (CD27-IgD-), as published previously[55,56].

### Preparation and sequencing of libraries: scRNA-seq, scTCR/BCR-seq, and GoT
Total PBMCs from VEXAS patients and healthy donors were subjected to scRNA-seq analysis using the Chromium Next GEM Single Cell 5′ Reagent Kits v2 (Dual Index) System (#PN-1000244, 10x Genomics), following the manufacturer's protocols. In brief, PBMCs were washed with 1X PBS with 0.04% FBS. Cell concentrations and viabilities were determined using the LUNA-II™ Automated Cell Counter (Logos biosystems) and the trypan blue staining method. Cells loaded into a Chip K were subjected to single cell partitioning, lysis, and barcoding using the Chromium Single Cell Controller (10x Genomics). Subsequently, cDNA was generated in a thermal cycler and purified with Dynabeads MyOne SILANE (#PN-2000048, 10x Genomics), followed by cDNA amplification. cDNA was allocated for gene expression library construction (50 ng of cDNA), targeted genotyping (10 µl of cDNA), and TCR/BCR library construction (2 µl of cDNA). Any remaining cDNA was stored. For gene expression library construction, amplified cDNA was fragmented, end-repaired, and A-tailing double-sided size-selected with solid phase reversible immobilization (SPRI)-select beads (#B23318, Beckman Coulter). For TCR/BCR library construction, V(D)J amplification was performed twice, followed by double-sided size selection with SPRIselect beads. For GoT library construction, three serial PCRs were performed on 10 µl of the remaining samples set aside during step 2.3, based on the *UBA1 M41* mutation of interest. A first PCR was performed with a

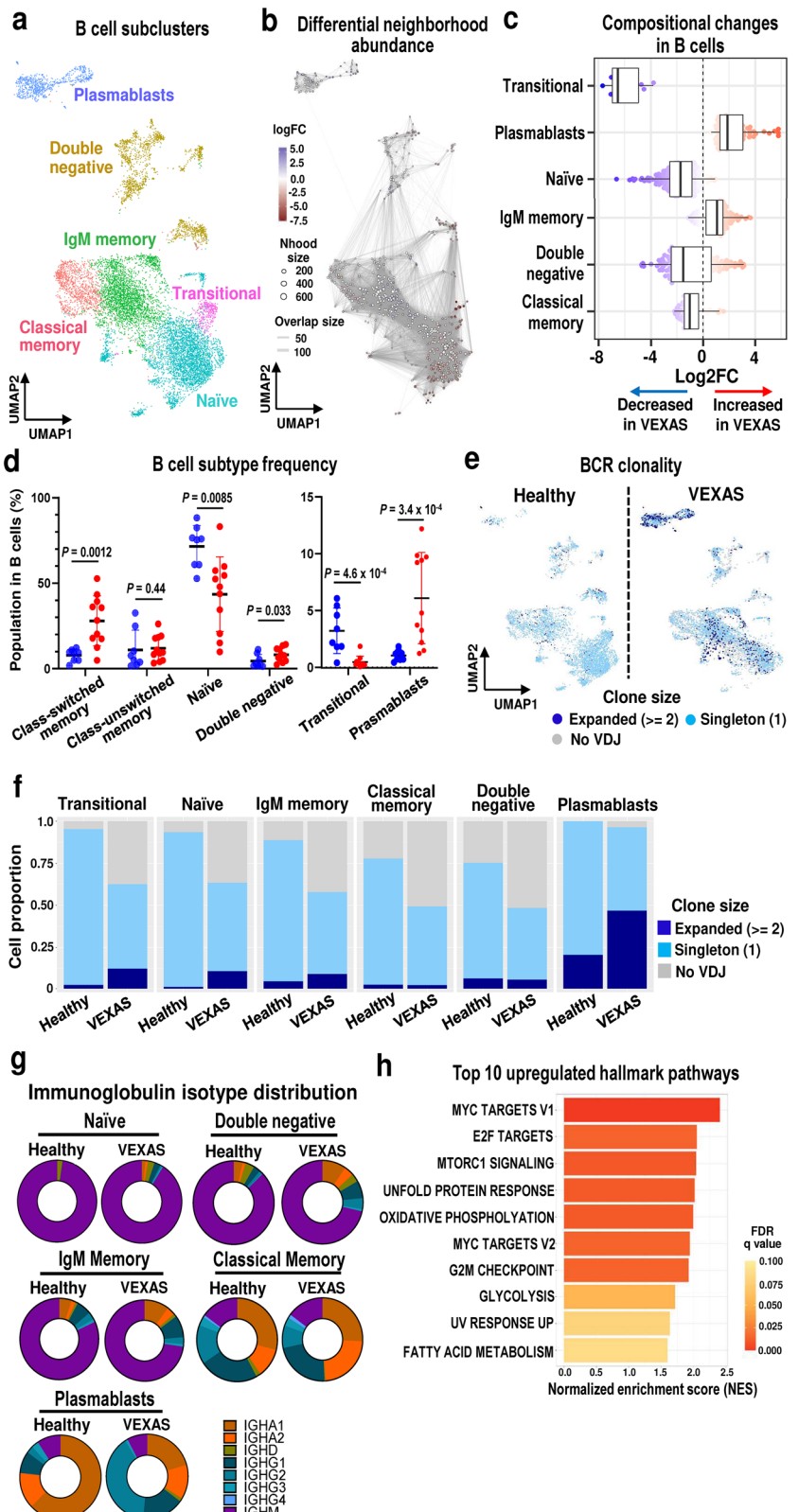

partial read 1 primer which binds to the partial read 1 of the Illumina sequencing handle (5'-CTACACGACGCTCTTCCGATCT-3') and a nested reverse primer (5'-GTCATGTAGGGTAACAGCCTTGAC-3') to amplify genotyping fragments before sample indexing using a thermocycler program: 95 °C for 2 min; 12 cycles of 95 °C for 30 s, 52 °C for 50 s, and 72 °C for 60 s; and 72 °C for 7 min. Using the 1st PCR product, a 2nd PCR was carried out with a forward PCR primer

(containing P5, sample index (i5), and a partial read 1 handle (5'-AATGATACGGCGACCACCGAGATCTACACXXXXXXXXXXXXACACTCTT TCCCTACACGACGCTC-3') to complete the read 1) and a second locus-specific reverse primer (containing a partial read 2 handle and a locus-specific region (5'-CGTGTGCTCTTCCGATCTCAACACATA-CAGCTGCCGGGAGTAAAGG-3') to *UBA1 M41* site): 95 °C for 2 min; 10 cycles of 95 °C for 30 s, 52 °C for 50 s, 72 °C for 60 s; and 72 °C for

**Fig. 6 | Skewed differentiation of B cells and clonal expansion of plasmablasts in VEXAS. a** A UMAP plot of 11,335 B cells. **b** A neighborhood (Nhood) graph of B cells, which was generated similarly as in Fig. 2a. **c** Beeswarm and box plots of each B cell subtype for VEXAS (n = 9) and healthy donors (n = 5), which were generated similarly as in Fig. 2b. **d** Proportions of B cell subtypes relative to the total number of B cells in VEXAS (n = 11, red dots) and healthy donors (n = 8, blue dots) by flow cytometry. Data are presented as mean with SD. *P* values were calculated with the two-sided unpaired Mann–Whitney *U* test. **e** UMAP plots of B cells from VEXAS patients (right) and healthy donors (left) colored by clonal expansion sizes.

**f** Distributions of B cell clone states in each subtype in VEXAS patients and healthy donors. **g** Isotype distributions of immunoglobulins expressed by each transcriptionally defined B cell subpopulation within all subjects. **h** A bar chart showing gene set enrichment scores of top 10 upregulated hallmark pathways in plasmablasts from VEXAS patients. A horizontal axis displays normalized enrichment score (NES) differences between VEXAS patients and healthy donors. A color scale indicates FDR values. BCR B-cell receptor. Source data are provided as a Source Data file.

7 min. After these two rounds of amplification and then SPRI purification, the 2nd PCR product was subjected to a 3rd PCR with a generic forward PCR primer (P5-generic, 5′-AATGATACGG CGACCACCGAGATCTACAC-3′) to retain the cell barcode (CB) and unique molecular identifier (UMI) together with an RPI primer (5′-CAAGCAGAAGACGGCATACGAGATXXXXXXXXXXXGTGACTGGAGTT CAGACGTGTGCTCTTCCGATCT-3′) to complete read 2 and P7 end of the library: 95 °C for 2 min; 6 cycles of 95 °C for 30 s, 52 °C for 50 s, 72 °C for 60 s; and 72 °C for 7 min. Qualities and quantities of the libraries were assessed using the Agilent 2100 Bioanalyzer (Agilent Technologies). Gene expression and TCR/BCR libraries were pooled together to receive around 20,000 and 5000 reads per cell, respectively, and sequenced using the Illumina NovaSeq 6000 system with read lengths of 26 bp read 1, 10 bp i7 index, 10 bp i5 index, 90 bp read 2. GoT libraries were sequenced separately to receive around 5000 reads per cell using the Illumina NovaSeq 6000 system with read lengths of 26 bp read 1, 10 bp i7 index, 10 bp i5 index, 90 bp read 2.

### Raw data preprocessing and quality control of scRNA-seq data

After single-cell libraries were sequenced using the Illumina system, the Cell Ranger ver 7.0.1 pipeline (https://www.10xgenomics.com/support/software/cell-ranger/latest) was utilized to process scRNA-seq raw data for read alignment to the genome and generation of gene-cell expression matrices[57]. Specifically, sequencing reads in FASTQ files were aligned to the human reference genome (hg38) using the STAR aligner[58] with annotation by ENSEMBL. The uniquely aligned reads were subjected to measurement of gene expression levels for all ENSEMBL genes with UMIs. Low quality cells were excluded from further analyses if the number of genes detected was < 500 genes/cell (potential fragments), >6000 genes/cell (potential doublets), or > 5% mitochondrial reads/cell, and remaining single cells were subjected to subsequent data analysis.

TCR reads were aligned to the GRCh38 reference genome and consensus TCR was annotated with the cellranger vdj program (10x Genomics, version 7.0.1). TCR libraries were sequenced with a final average of 7827 read pairs/cell. On average, 5697 reads mapped to either the *TRA* or *TRB* loci in each cell. The 10x cellranger vdj pipeline provided at https://support.10xgenomics.com/single-cell-vdj/software/pipelines/latest/using/vdj was used to perform TCR annotation. Barcodes with higher numbers of Unique UMI counts than those of simulated background were considered as cell barcodes. V(D)J read filtering and assembly were implemented as described in a previous study[59]. cellranger trimmed known adaptor and primer sequences from the 5′ and 3′ ends of reads, followed by filtering away reads lacking at least one 15-bp exact match against at least one reference segment (TCR, TRA, and TRB gene annotations in ENSEMBL version 87). Subsequently, cellranger built a De Bruijn graph of reads independently, resulting in de novo assembly for each barcode. The assembler produced contig sequences assigned at least one UMI and each assembled contig was aligned against all of the germline segment reference sequences of the V, D, J, C, and 5′ UTR regions. cellranger searched a CDR3 motif (Cys-to-FGXG/WGXG) in a frame defined by the start codon in the L + V region or all 6 frames when the L + V region was absent. Most cell barcodes contained two matching productive contigs, comprising either a *TCRA* or a *TCRB*.

However, there was a biological possibility that few productive contigs (low sensitivity) or > 2 productive contigs (some cells contain more than one TCRB or TCRA chain) were associated with one cell barcode[60]. Similarly, the cellranger vdj program also processed BCR reads with the IMGT database of GRCh38 genome as reference. Only productive contigs of BCR were kept for further analyses.

Downstream analyses were mainly performed using the R software package[61] in Seurat[62] (http://satijalab.org/seurat/, v4.0.4) on PBMCs[63]. Raw reads in each cell were first scaled by a library size to 10,000 and then log-transformed. To improve downstream dimensionality reduction and clustering, we selected top 2000 highly variable genes for PCA of high-dimensional data. Top 50 principal components were imported into function of FindIntegrationAnchors to integrate datasets from all samples and the integrated data were used for unsupervised clustering of cells with a graph-based clustering approach[64], and further reduced dimension with UMAP. Cell type identity was assigned to each cluster based on significance in overlap between signature genes of PBMCs[12] and cluster-specific genes (Fisher's exact test). Gene Set Enrichment Analysis (GSEA; http://software.broadinstitute.org/gsea) and Gene Ontology (GO)[65] (the Gene Ontology Consortium, 2023) were used to interpret gene set enrichment and pathways of defined differentially expressed genes.

### GoT bioinformatics analysis

We downloaded the IronThrone which genotypes individual cells (https://github.com/dan-landau/IronThrone-GoT), and modified the code to adapt to our sequencing data and the slurm high performance computing system. In brief, all amplicon reads were shuffled and subsetted into smaller groups of reads (default 125,000 reads per group). The original IronThrone algorithm was then run on each one of these groups in parallel. Pairwise Levenshtein distances between all UMIs paired were calculated within a single-cell barcode, and UMI pairs with a Levenshtein distance below a predetermined threshold [default = ceiling (0.1 × UMI length), or 2 bases for a 12-base UMI] were defined as 'matches'. The initial UMI was obtained by the greatest number of matched UMIs. This process was repeated for the UMI with the next highest number of matches until no additional collapsing was possible. Rare UMIs with an equal number of mutant and wild-type reads were removed as ambiguous. Given the hemizygous nature of the *UBA1* mutation in male patients, cell barcodes with a minimum of one wild-type read were genotyped as wt*UBA1*, while those with a minimum of one mutant read were genotyped as mt*UBA1*. The code is available at https://github.com/shouguog/NHLBIGoT.

### Cell type assignment in major cell types

To identify clusters within each major cell type, we performed a second round of clustering and cell type annotation on monocytes/DCs, NK cells, T cells (CD4+ T, CD8+ T, MAIT, and gdT), and B cells (B and plasmablasts), separately.

For cell type assignment of monocytes, the gene expression data of monocytes (GSE25913) were downloaded[15], which include the classical (CD14++CD16−), intermediate (CD14++CD16+), and nonclassical (CD14+CD16++) monocyte subtypes. Subtype specific genes were designated as the highest expressed genes in one subtype on the rest. A monocyte subtype was classified by assigning to each cluster based

on significance of overlapping between monocyte subtypes and cluster-specific genes (Fisher's exact test).

For cell type assignment of NK cells, the gene expression data of NK cells (GSE197037) were downloaded[21], which include the CD56[bright] NK, Early CD56[dim] NK, CD56[dim] NK, and adaptive-like NK cell subtypes. Subtype specific genes were designated as the highest expressed genes in one subtype on the rest. A NK cell subtype was classified by assigning to each cluster based on significance of overlapping between NK subtypes and cluster-specific genes.

For cell type assignment of T cells, the gene expression data of T cells (GSE93777) were downloaded[23], which include naïve, central, and effector T cell populations. Top 250 most population-specific genes were as signatures of subtypes. We used this gene set to define cell types. CD4[+], CD8[+], and related subtypes were assigned to each cluster based on significance in overlap between T cells and cluster-specific genes.

For cell type assignment of B cells, the gene expression data of B cells (E-MTAB-9544) were downloaded[25], which include the transitional B, naïve B, IgM memory B, classical memory B, and double negative B cell subtypes. Subtype specific genes were designated as the highest expressed genes in one subtype on the rest. A B cell subtype was classified by assigning to each cluster based on the significance of overlapping between B cell subtypes and cluster-specific genes.

We then manually refined all cell cluster annotations based on the expression of canonical marker genes (Supplementary Fig. 3a, 5a, 6a, and 7a).

### Differential abundance analysis

Differences in cell abundances between the baseline samples of patients and the healthy donors were analyzed by differential abundance testing with the miloR package (https://bioconductor.org/packages/release/bioc/html/miloR.html)[66]. Specifically, the Seurat objects were converted to SingleCellExperiment objects, and the PCAs and UMAPs of Seurat objects were also assigned to the single CellExperiment objects. Each neighborhood was assigned to a cell-type label based on the majority voting of cells belonging to that neighborhood. A neighborhood was labeled as "Mixed" if the most abundant label was present in < 75% of cells within that neighborhood[66].

### Single-cell mutation identification and analysis using cb_sniffer

cb_sniffer (https://github.com/sridnona/cb_sniffer), a Pysam-based tool, was used with default parameters[14] to identify single-nucleotide variations in *UBA1* in single cells from aligned sequence data generated by cellranger. Reads that had no Chromium Cellular Barcode (CB) tag or no Chromium Molecular Barcode (UB) tag were filtered out. Then, cell-associated tags for downstream analyses of UMIs were obtained. Usually, duplicate reads existed for a given UB and a base at a mutant position were identical across all reads. In rare cases when there were inconsistent reads, the most common base was chosen if a mutation was present in at least 75% of the reads. All reads corresponding to the UB were discarded when there was no common base at the mutation positions (> 75% reads).

### Differentially expressed genes and heatmap generation

Differentially expressed genes of VEXAS patients' and healthy donors' cells were defined using FindMarkers function in Seurat in which a gene expression level in one cell subset was compared with those in all others. Genes with *P* value < 0.05 and Log (average fold change) > 0.1 were regarded as differentially expressed genes. Heatmaps and network visualization were created with ggplot2 and heatmap.2 in the R package.

### Gene set enrichment analysis

Preranked gene-set analysis on MsigDB v2023.2 Hallmark gene sets was performed using preranked gene lists with the GSEA software (http://software.broadinstitute.org/gsea). Genes were preranked according to log2 fold change values for all preranked gene-set analysis procedures.

### Definition of module scores

TNF response scores were calculated using a gene set termed 'GOBP_RESPONSE_TO_TUMOR_NECROSIS_FACTOR1'(GO:0034612). IFN-α response scores were calculated using a gene set termed 'GOBP_RESPONSE_TO_TYPE_I_INTERFERON'(GO:0034340). IFN-γ response scores were calculated using a gene set termed 'GOBP_RESPONSE_TO_INTERFERON_GAMMA'(GO:0034341). Cytotoxicity scores of NK cells were calculated using a gene set termed 'GOBP_LEUKOCYTE_MEDIATED_CYTOTOXICITY' (GO:0001909). Apoptosis scores were calculated using a gene set termed 'HALLMARK_APOPTOSIS'. Cytotoxicity scores of CD8[+] T cells were calculated using a gene set termed 'GOBP_T_CELL_MEDIATED_CYTOTOXICITY'(GO:0001913).

We also calculated an HLA class II gene score, sepsis-associated monocyte signature score, and exhaustion score based on the following gene sets. The HLA class II genes consist of *HLA-DRA*, *HLA-DRB1*, *HLA-DRB5*, *HLA-DQA1*, *HLA-DQA2*, *HLA-DQB1*, *HLA-DMA*, *HLA-DMB*, *HLA-DPA1*, and *HLA-DPB1*. The alarmin-related S100A genes consist of *S100A8*, *S100A9*, and *S100A12*. The Exhaustion genes consist of *CXCL13*, *HAVCR2*, *PDCD1*, *TIGIT*, *LAG3*, *CTLA4*, *LAYN*, *RBPJ*, *VCAM1*, *GZMB*, *TOX*, and *MYO7A*. The sepsis-associated monocyte signature genes were defined based on the published reference gene set[16]. The module scores were calculated with AddModuleScore function built in the Seurat[62].

### Comparison of gene pathway module scores

To compare dysfunctional and inflammatory gene profiles in monocytes of VEXAS with several other rheumatological diseases, we calculated module scores (expression levels) of several pathways in monocytes of VEXAS patients in the current study, with data from published datasets (GSE135779) for systemic lupus erythematosus (SLE), E-GEAD-635 for microscopic polyangiitis (MPA) patients, and GSE198616 for Behcet's disease (BD) patients. Gene sets for the interferon response pathways, the TNF response pathway, the HLA class II genes, the S100A genes, and the sepsis-associated monocyte signature genes were as described above. Their module scores (expression levels) in our and these three datasets were calculated with the AddModuleScore function. The module scores were normalized with healthy donors included in individual studies, and the double-sided t-test was used to assess the difference between VEXAS and three other diseases (SLE, MPA, and BD).

### Dynamic changes of mt*UBA1* and wt*UBA1* cells in VEXAS patients along differentiation

In order to estimate T cell differentiation, trajectory inference was performed with the R package Slingshot[67]. The analyses were performed for CD8[+] T cells. The UMAP matrix was fed into Slingshot and a naïve CD8[+] T cell population was manually designated as the root of all inferred trajectories considering naïve CD8[+] T cells differentiate to other CD8[+] T cells. A pseudotime variable was inferred by fitting simultaneous principal curves for further analysis. Similarly, pseudotime was estimated by Slingshot in monocytes and NK cells.

### Ligand receptor analysis

Cell-cell interactions based on the expression of known ligand-receptor pairs in different cell types were calculated using Connectome version 1.0.2[68]. The algorithm was run on log-normalized expression values for cell populations of PBMCs with default parameters and no subsampling to identify enriched ligand-receptor pairs in VEXAS patients and healthy donors.

## Diversity index calculation

Several methods represent the number of clones (identical TCR chains) present (richness) and of their relative frequency (evenness). The Shannon entropy weighs both of these aspects of diversity equally, and it is an intuitive measure whereby the maximum value is determined by the total size of the repertoire. Entropy values decrease with increasing inequality of frequency as a result of clonal expansion. The Shannon entropy in a population of N clones with nucleotide frequency *pi* is defined by Eq. (1):

$$H(P) = -\sum_{i=1}^{n} p_i \log_2 p_i \tag{1}$$

The Gini coefficient is a number aimed at measuring the inequality in a distribution. It is most often used in economics to measure a country's wealth distribution and has been widely used in the diversity assessment of TCRs. The Gini index and Shannon entropy for diversity and clonality analyses were calculated with the R package of tCR (https://imminfo.github.io/tcr/)[69].

## Identification of TCR motifs with shared antigen specificity using GLIPH2

GLIPH2[24] was applied to T cells of VEXAS patients and healthy donors to identify clusters of TCRs that recognize the same epitope based on CDR3β amino acid sequence similarities, with default parameters. CDR3β amino acid sequences of top 1000 most abundant CDRs were used to identify significant motif lists and associated TCR convergence groups.

## Somatic hypermutation analysis

The BCR sequence data was processed by the Immcantation toolbox (v4.0.0)[70] using the IgBLAST and IMGT germline sequence databases, with default parameter values. Specifically, the output files filtered_contig.fasta and filtered_contig_annotations.csv from cellrangerVDJ were input into AssignGenes.py and MakeDb.py to create a tab-delimited database file to store sequence alignment information. Then, the number of somatic mutations for each sequence was calculated using observedMutations (Shazam v1.1.0). The mutation rate was categorized into four groups: germline (mutation frequency = 0), low (0 < mutation frequency ≤ 3 %), and high (mutation frequency > 3%). Antibody isotypes for cells were identified based on the presence of immunoglobulin constant region categories. Paired scBCR-seq data were integrated with scRNA-seq based on their matched unique cell barcodes.

## Statistics

No data were excluded from the analyses. No statistical method was used to predetermine sample size. We did not use any study design that required randomization or blinding. Statistical analyses were performed as described in the figure legends. Comparison between groups was performed using the GraphPad Prism (v10.2.0; GraphPad software, La Jolla, CA).

## Reporting summary

Further information on research design is available in the Nature Portfolio Reporting Summary linked to this article.

## Data availability

The raw and analyzed sequencing data in this study have been deposited in the NCBI Gene Expression Omnibus under the primary accession code GSE249131 and Sequence Read Archive under accession code PRJNA1047528, and are publicly available. All data are included in the Supplementary Information or available from the authors, as are unique reagents used in this Article. The raw numbers for charts and graphs are available in the Source Data file whenever possible. Source data are provided with this paper.

## Code availability

Code for Genotyping of Transcriptomes is available at a dedicated Github repository [https://github.com/shouguog/NHLBIGoT] and Zenoda [https://doi.org/10.5281/zenodo.15046442][71]. Any additional analysis scripts and information required to reanalyze the data reported in this paper are available from the lead contact upon request.

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

## Acknowledgements
We thank Tania Machado (NHLBI/NIH), Olga Rios (NHLBI/NIH), and Katherine Roskom (NHLBI/NIH) for assistance in obtaining samples, and patients and healthy volunteers who donated blood. Sequencing and technical supports were provided by the DNA Sequencing and Genomics Core of NHLBI/NIH. FACS was provided by the flow cytometry core of NHLBI/NIH. O-link was provided by the Clinical Support Laboratory of NCI/NIH. Our scripts for genotyping analysis are based on IronThrone-GoT from Dr. Dan A Landau' s group at Weill Cornell Medical College. This research was supported by the Intramural Research Program of the NHLBI.

## Author contributions
H.M. designed and performed the experiments, analyzed data, and wrote the manuscript. S.G. did bioinformatics analysis and wrote the manuscript. Z.W., F.G.R., M.B., X.F., L.A., D.Q.R., and S.K. performed experiments. H.L. did bioinformatics analysis. E.M.G., I.D., P.C.G., N.S.Y., and B.P. provided patient care. F.G.R wrote and edited the manuscript. S.K. and G.F. supervised data analysis and edited the manuscript. N.S.Y. and B.P. conceived, designed, and supervised the experiments, analyzed results, and edited the manuscript. All authors reviewed and approved the final version.

## Funding

## Competing interests
The authors declare no competing interests.
