## [Transparent Peer Review file · Nature Communications]

In depth transcriptomic profiling defines a landscape of dysfunctional immune responses in patients with VEXAS syndrome

Corresponding Author: Dr Zhijie Wu

Version 0:

Reviewer comments:

Reviewer #2

(Remarks to the Author)
Comments

In this study, Mizukami et al. have performed multi-omics single-cell RNA analysis, cytokine multiplex assays and in vitro functional assays using peripheral blood samples derived from VEXAS patients. Authors have demonstrated a broad immune system activation with upregulation of multiple inflammatory response pathways and proinflammatory cytokines. Functional assays revealed that monocytes derived from VEXAS patients exhibit impaired efferocytosis and blunted cytokine production. Among lymphocyte fractions which are predominantly UBA1 wild-type, authors have detected clonal expansion of effector memory CD8+ T cells and plasmablasts at the expense of transitional B cells. Although the conclusions are interesting in the field and potentially could be clinically important, there are weaknesses and limitations that need to be addressed.

Major comments

1. In figures 3D and 3f, authors have shown upregulated alarmin-related S100A genes and increases in sepsis-associated monocyte signature genes in VEXAS monocytes. However, authors should clarify if these transcriptional profiles of VEXAS monocytes are independent of patients' clinical course, especially infectious/septic complications.
2. If upregulated alarmin-related S100A genes and increases in sepsis-associated monocyte signature genes in VEXAS monocytes are independent of patients' clinical course, authors are recommended to describe a potential cause for this observation. (related to major comments #1)
3. Functional assays revealed that monocytes derived from VEXAS patients exhibit blunted cytokine production. Nonetheless, VEXAS monocytes still showed upregulation of inflammatory response pathways. This seems to be paradoxical to this reviewer. Authors are recommended to describe a potential cause for this discrepancy.
4. If authors believe that TLR ligands other than LPS may affect differently to VEXAS monocytes which might be one of the molecular bases for the upregulation of inflammatory response in these monocytes, authors should repeat functional assays using other TLR ligands. (R848 and Pam3CSK4, etc.)
5. In figure 3e, authors have shown upregulated apoptosis pathways in VEXAS monocytes and mDCs. Nevertheless, a ratio of mtUBA1 monocytes along pseudotime trajectories was similar to that of wtUBA1 monocytes as shown in figure 3h, which is consistent with the notion that VEXAS syndrome is a clonal myeloid disease. This again seems to be paradoxical to this reviewer. (If apoptosis is a common feature of VEXAS monocytes and mDSs, these cell fractions should eventually contract) Authors are recommended to describe a potential cause for this discrepancy.
6. In discussion, authors have stated that monocytes may be involved with a secondary activation of pro-inflammatory pathways in other immune cells through upregulated ligand-receptor interactions between these cells, despite dysfunctional features of VEXAS monocytes. This again seems to be paradoxical to this reviewer. (If VEXAS monocytes are dysfunctional, how can these cells upregulate ligand-receptor interactions?) Authors are recommended to describe a potential cause for

this discrepancy.

Minor comments

1. In page 6, line 156, "was performed to compared" should be "was performed to compare". (remove "d" from "compared")

Reviewer #3

(Remarks to the Author)

The authors conducted a comprehensive single-cell analysis of VEXAS syndrome using advanced methods, including scRNA-seq and scTCR/BCR-seq, yielding significant insights into this poorly characterized disease. Their findings, such as the expansion of plasmablasts and RNA expression differences in mtUBA1 and wtUBA1 cells, provide valuable contributions to understanding VEXAS. Notably, the observed defects in monocyte efferocytosis and antigen-independent B cell lymphopoiesis stand out as intriguing and novel. However, some areas lack sufficient detail. Addressing these gaps could enhance the study's clarity and its potential to inform therapeutic development.

Major comments:

1. Gender Bias in Sample Selection (Line 96-98):

The authors included only male samples in their study. While this choice might have been made to reduce variability or due to the predominance of VEXAS in males, it introduces a significant gender bias and limits the applicability of findings to female patients. The manuscript should explicitly justify this decision and discuss the potential impact of gender exclusion on the study's conclusions.

2. Clarification of "N/A" Notation in UBA1 Variant (Protein) Column (Table 1):

In Table 1, the "N/A" notation in the UBA1 variant (protein) column for a patient in the scRNA-seq cohort requires clarification. The authors should specify whether this indicates the absence of a detected UBA1 variant or missing data. If no amino acid sequence variation was observed in this patient, were additional investigations conducted to clarify why the patient was classified as having VEXAS? Furthermore, if the diagnosis was based on a genetic sequence mutation rather than an amino acid sequence variation, the inclusion of this patient's data (Cohort 16) in the GoT library seems necessary. If there were specific reasons for excluding this data, they should be explicitly explained.

3. Discrepancy Between scRNA-Seq and Flow Cytometry Data (Line 192-201):

The authors identified elevated adaptive-like NK cells in VEXAS patients using scRNA-seq, characterized by high expression of CD3E, LAG3, and IL-32, with no expression of FCER1G and TCRs. However, the observed discrepancy between scRNA-seq and flow cytometry results raises concerns. Specifically, some adaptive-like NK cells lack canonical markers such as KLRC2/NKG2C, which are typically associated with HCMV-induced adaptive NK cells. To address the potential bias introduced by the HCMV-positive VEXAS patient (VEXAS17), it would strengthen the study to include additional experiments using surface markers that are not exclusively linked to HCMV-driven adaptive NK cells. This would help validate the existence of adaptive-like NK cells in VEXAS patients arising from other triggers and confirm their role in the disease.

4. Clonal Expansion in T Cells (Line 214-247):

The clonal expansion of CD8+ T cells with restricted TCR usage is an intriguing finding. However, the lack of shared target antigens or disease-specific TCR clusters raises questions about the drivers of clonal expansion. The authors should discuss whether this phenomenon might reflect non-specific inflammation or neoantigen-driven responses resulting from mtUBA1-related dysregulation in myeloid cells.

5. Broader Implications of Protein Processing in B Cells (Line 280-285):

The significant upregulation of genes related to ER stress and protein processing in plasmablasts suggests enhanced antibody production in VEXAS. Notably, VEXAS patients exhibited increased somatic hypermutation (SHM) in plasmablasts, alongside shifts in immunoglobulin isotype usage, transitioning from IgA dominance observed in healthy donors to IgG predominance in patients. The authors could further explore whether these changes in SHM and immunoglobulin usage contribute to the production of autoantibodies and how they correlate with disease severity and progression.

6. Lack of Healthy Donor GoT Data and Proportion-Dependent Changes (Supplementary Table 2, Line 111-114, Line 202-212):

In Supplementary Table 2, only GoT data from VEXAS patients are presented. It is unclear whether GoT data were not generated for healthy donor samples. If GoT data were exclusively generated for VEXAS patients, this should be explicitly stated in the manuscript for clarity. Additionally, NK cells are described as being directly impacted by mtUBA1, but it would strengthen the study to include data on how the characteristics of NK cells change with varying proportions of mtUBA1. If similar proportion-dependent changes are observed in other cell types, those findings should also be included to provide a broader understanding of mtUBA1's impact. Lastly, adding plots that depict the ratios of mtUBA1 to wtUBA1 across cell types would enhance data visualization and help readers better interpret the results.

Minor comments:

1. Authors should cite the full name of UPR. (Line 73)

2. The naming of DC subtypes alternates between "cDC" and "mDC" throughout the manuscript. It would be preferable to

use a consistent term. If "mDC" is chosen, the full name should be specified for clarity (e.g., in Line 142, Line 148, Figure 3a, and Figure 3c).

3. The term "nctMono" appears to be a typo and should be corrected to "ncMono." If this is not a typo, the full name should be provided, and the authors should explain why this new cell type, not mentioned in previous annotations, is being introduced. (Figure 3e)

Version 1:

Reviewer comments:

Reviewer #2

(Remarks to the Author)

The authors have done an outstanding job and thoroughly and convincingly addressed all points raised by the reviewers. I have no further comments.

Reviewer #3

(Remarks to the Author)

I have carefully reviewed the authors' responses to my initial comments, and I find their revisions and justifications to be appropriate and well-supported. Below is my final assessment of their responses.

1. Gender Bias in Sample Selection

The authors have adequately addressed the concern regarding the exclusion of female patients by referencing relevant literature and explicitly discussing this limitation in the manuscript. Their explanation regarding the rarity of female VEXAS patients due to the X-linked nature of the mutation is well-supported.

2. Clarification of "N/A" Notation in UBA1 Variant Column

The authors have provided a clear explanation regarding the splice-site mutation (c.118-1G>C) and its impact on protein translation, which justifies why "N/A" was originally noted. The removal of "N/A" and the addition of clarifying text improve the manuscript's clarity. Their reasoning for excluding VEXAS16 from the GoT library is scientifically sound and now well-articulated.

3. Discrepancy Between scRNA-Seq and Flow Cytometry Data

The authors have strengthened their findings by conducting additional flow cytometry experiments with an expanded patient cohort. The new data confirms the scRNA-seq findings and eliminates the previous concern regarding a potential bias from a single HCMV-positive patient. The revised Supplementary Figures and added references further support their claims.

4. Clonal Expansion in T Cells

The authors have provided a well-reasoned discussion on how excessive non-specific antigens from mtUBA1 myeloid cells could drive CD8+ T cell clonal expansion. Their revised discussion section appropriately reflects this hypothesis.

5. Broader Implications of Protein Processing in B Cells

While the role of autoantibodies in VEXAS remains uncertain, the authors have adequately contextualized their findings by referencing autoimmune diseases where plasmablast expansion correlates with autoantibody production. Their discussion now highlights the need for future large-cohort studies to explore this potential link.

6. Lack of Healthy Donor GoT Data and Proportion-Dependent Changes

The authors have clarified why GoT data were not generated for healthy donors, making this explicit in the revised manuscript. Additionally, their newly added Supplementary Figure 5f effectively illustrates how inflammatory and apoptotic pathway gene module scores correlate with the proportion of mtUBA1 cells, further strengthening the mechanistic insights of the study. The adjustments to Figure 1f also improve data interpretation.

7. Minor Comments

The manuscript has been revised to address all minor concerns, including terminology inconsistencies, typographical errors, and missing full names of abbreviations. These changes improve the overall clarity and professionalism of the manuscript.

Final Assessment:

The authors have thoroughly and satisfactorily addressed all concerns raised in my initial review. The additional experiments, clarifications, and revisions enhance the robustness and clarity of their findings. I find the manuscript to be well-revised and scientifically sound, and I have no further requests for modification.

RESPONSE TO REVIEWERS' COMMENTS

Reviewer #2 (Remarks to the Author)

Comments

In this study, Mizukami et al. have performed multi-omics single-cell RNA analysis, cytokine multiplex assays and in vitro functional assays using peripheral blood samples derived from VEXAS patients. Authors have demonstrated a broad immune system activation with upregulation of multiple inflammatory response pathways and proinflammatory cytokines. Functional assays revealed that monocytes derived from VEXAS patients exhibit impaired efferocytosis and blunted cytokine production. Among lymphocyte fractions which are predominantly UBA1 wild-type, authors have detected clonal expansion of effector memory CD8+ T cells and plasmablasts at the expense of transitional B cells. Although the conclusions are interesting in the field and potentially could be clinically important, there are weaknesses and limitations that need to be addressed.

Major comments

1. In figures 3D and 3f, authors have shown upregulated alarmin-related S100A genes and increases in sepsis-associated monocyte signature genes in VEXAS monocytes. However, authors should clarify if these transcriptional profiles of VEXAS monocytes are independent of patients' clinical course, especially infectious/septic complications.

Response: One patient (VEXAS 13) had bacterial infections (bacteremia and cellulitis) one month before sampling, which was well-controlled and inactive at the time of blood collection. Other patients were free of infectious complications at sampling. VEXAS 13 did not show higher score of sepsis-associated monocyte signature (-0.00499) than did the other cases (mean -0.00224 [range, from -0.014670 to 0.011562]), and dysfunctional monocyte gene signatures thus appeared independent of infectious complication. For clarification, we have added the following text on page 7, line 157-160 of the revised manuscript.

"Although none of the patients had active bacterial infection at the time of blood collection, the gene expression profile of monocytes from VEXAS (increased alarmin-related S100A gene and decreased HLA class II gene expression) was similar to dysfunctional monocytes that have been described in severe sepsis."

2. If upregulated alarmin-related S100A genes and increases in sepsis-associated monocyte signature genes in VEXAS monocytes are independent of patients' clinical course, authors are recommended to describe a potential cause for this observation. (related to major comments #1)

Response: Most of the patients in our scRNA-seq cohort were in late phase of their disease (median time from symptom onset was 7.1 [range, from 1.9 to 10.1] years). Prolonged exposure to inflammatory cytokines from dysregulated immune cells may have caused immune paralysis despite global upregulation of inflammatory response pathways. Many groups have reported diminished capacity of septic patients' monocytes to release proinflammatory cytokines in vitro in response to additional exposure to LPS, TLR agonists, or various bacterial compounds even after acquisition of immune activation.¹

We now have added the following text on page 14, line 330-336 of the revised manuscript.

“Most of the patients in the scRNA-seq cohort were in later phase of their disease (median time from symptom onset was 7.1 [range, from 1.9 to 10.1] years), and prolonged inflammation itself may have diminished the capacity of VEXAS monocytes to release proinflammatory cytokines in response to additional exposure to stimuli, despite intrinsically upregulated immune response pathways.”

3. Functional assays revealed that monocytes derived from VEXAS patients exhibit blunted cytokine production. Nonetheless, VEXAS monocytes still showed upregulation of inflammatory response pathways. This seems to be paradoxical to this reviewer. Authors are recommended to describe a potential cause for this discrepancy.

Response: Please see our response to major comment #2.

4. If authors believe that TLR ligands other than LPS may affect differently to VEXAS monocytes which might be one of the molecular bases for the upregulation of inflammatory response in these monocytes, authors should repeat functional assays using other TLR ligands. (R848 and Pam3CSK4, etc.)

Response: Based on the reviewer’s suggestion, we collected blood samples from additional 4 VEXAS patients and 5 healthy donors to perform in vitro experiments using other TLR ligands. VEXAS monocytes did not respond to other TLR ligands including R848 and Pam3CSK4 as well as to LPS, again demonstrating loss of effective response to external stimuli by VEXAS monocytes.

We now have deleted the original text in page 16, line 395-400. Further, we have added new Supplementary Fig. 4d, and the following text in page 8, line 193-195 of the revised manuscript.

Original manuscript:

“Third, LPS was used for stimulation of monocytes in this study, but other TLR ligands such as R848 (a synthetic ligand for TLR7/8) and Pam3CSK4 (a synthetic ligand for TLR1/2) were not tested due to limited number of cells. Monocytes from severe COVID-19 patients are reported to show impaired cytokine production upon stimulation through LPS and R848, but not Pam3CSK4. Alteration of cytokine production from VEXAS monocytes might also depend on the type of stimuli.”

-> deleted.

Revised manuscript:

“Blunted cytokine production by monocytes from VEXAS patients was also observed after stimulation with other toll-like receptor (TLR) ligands (Supplementary Fig. 4c),”

New Supplementary Fig. 4d

(Supplementary Fig. 4d) Cytokine detection of IL-6, IL-1 β , and TNF- α in culture supernatants of purified CD14⁺ monocytes (VEXAS patients, n = 4; healthy donors, n = 5) after 8 h in-vitro incubation with or without 100 ng/mL of lipopolysaccharide (LPS), 100 ng/mL of Pam3CSK4, and 2.5 μ M of R848. Data are presented as a mean value with SD. P values were calculated using the two-sided unpaired Mann-Whitney U test.

5. In figure 3e, authors have shown upregulated apoptosis pathways in VEXAS monocytes and mDCs. Nevertheless, a ratio of mtUBA1 monocytes along pseudotime trajectories was similar to that of wtUBA1 monocytes as shown in figure 3h, which is consistent with the notion that VEXAS syndrome is a clonal myeloid disease. This again seems to be paradoxical to this reviewer. (If apoptosis is a common feature of VEXAS monocytes and mDSs, these cell fractions should eventually contract) Authors are recommended to describe a potential cause for this discrepancy.

Response: As shown in Supplementary Table 1, absolute number of monocytes in each patient were low. Similar ratios of mtUBA1 and wtUBA1 monocytes to total VEXAS monocytes indicated that not only mtUBA1 monocytes but also wtUBA1 monocytes decreased in VEXAS patients. To clarify this point, we have created new Supplementary Fig. 3d in which we showed both wtUBA1 and mtUBA1 monocytes upregulated apoptosis pathway with differentiation.

New Supplementary Fig. 3d

(Supplementary Fig. 3d) Dynamic changes of gene module scores of the apoptosis pathway in wild-type UBA1 (wtUBA1) and UBA1-mutated (mtUBA1) monocytes from VEXAS patients and monocytes from healthy donors along differentiation. x axis, pseudotime ordering from classical monocytes to nonclassical monocytes estimated by slingshot; y axis, gene module scores of apoptosis pathway.

We also have added the following text in page 7, line 176-178 of the revised manuscript.

“Gene module scores of the apoptosis pathway in wt*UBA1* monocytes tracked by pseudotime were similar to those of mt*UBA1* monocytes, and both were higher than for monocytes from healthy donors (Supplementary Fig. 3d).”

In addition, we have created a new Fig. 1b and added the following text in page 4, line 103-105 of the revised manuscript to clarify monocytopenia in VEXAS patients.

“Consistent with previous reports, patients were cytopenic, with prominent macrocytic anemia, thrombocytopenia, monocytopenia, and lymphocytopenia (Fig. 1b)”.

New Fig. 1b

(Fig. 1b) Patients’ hemoglobin levels (HGB), platelet counts (PLT), white blood cell counts (WBC), neutrophil counts, monocyte counts, B-cell counts, NK-cell counts, and T-cell counts. Background shading shows a normal reference range for each parameter. Data are presented as mean value with SD.

6. In discussion, authors have stated that monocytes may be involved with a secondary activation of pro-inflammatory pathways in other immune cells through upregulated ligand-receptor interactions between these cells, despite dysfunctional features of VEXAS monocytes. This again seems to be paradoxical to this reviewer. (If VEXAS monocytes are dysfunctional, how can these cells upregulate ligand-receptor interactions?) Authors are recommended to describe a potential cause for this discrepancy.

Response: We agree with the reviewer that the paradoxically increased monocyte interactions with other cells and corresponding dysfunctional features is not entirely understood. Exhausted T cells have been shown to have diminished cytokine production and are functionally impaired but they still retain ligand-receptor interactions with other immune cells.² We speculate whether we are observing similar functional changes in VEXAS monocytes. However, as the reviewer pointed out, it remains unclear whether dysfunctional monocytes are still actively involved in autoinflammation in blood. Therefore, we have simplified and clarified our conclusion on page 14, line 334-336.

“Despite dysfunctional features, monocytes were highly interactive with other immune cells, as imputed computationally from ligand-receptor interactions.”

Minor comments

1. In page 6, line 156, “was performed to compared” should be “was performed to compare”. (remove “d” from “compared”).

Response: We have corrected this typographic error.

Reviewer #3 (Remarks to the Author):

The authors conducted a comprehensive single-cell analysis of VEXAS syndrome using advanced methods, including scRNA-seq and scTCR/BCR-seq, yielding significant insights into this poorly characterized disease. Their findings, such as the expansion of plasmablasts and RNA expression differences in mtUBA1 and wtUBA1 cells, provide valuable contributions to understanding VEXAS. Notably, the observed defects in monocyte efferocytosis and antigen-independent B cell lymphopoiesis stand out as intriguing and novel. However, some areas lack sufficient detail. Addressing these gaps could enhance the study's clarity and its potential to inform therapeutic development.

Major comments:

1. Gender Bias in Sample Selection (Line 96-98):

The authors included only male samples in their study. While this choice might have been made to reduce variability or due to the predominance of VEXAS in males, it introduces a significant gender bias and limits the applicability of findings to female patients. The manuscript should explicitly justify this decision and discuss the potential impact of gender exclusion on the study's conclusions.

Response: The genetic basis of VEXAS is an X-chromosome gene mutation, and female VEXAS patients are extremely rare, limited to individuals who have monosomy X.^{3,4} Although we are a large referral center, we have had no women referred to us. We have added new references ref. 50 (Stubbins RJ, *et al.* VEXAS syndrome in a female patient with constitutional 45,X (Turner syndrome). *Haematologica* 107, 1011-1013 (2022)) and ref. 51 (Echerbault R, Bourguiba R, Georgan-Lavialle S, Lavigne C, Ravaiau C, Lacombe V. Comparing clinical features between males and females with VEXAS syndrome: data from literature analysis of patient reports. *Rheumatology (Oxford)* 63, 2694-2700 (2024)) and the following text as a limitation of this study in page 17, line 410 - 412 of the revised manuscript.

“Third, female VEXAS patients are extremely rare, limited to individuals who have monosomy X, there is obvious sex bias.”

2. Clarification of "N/A" Notation in UBA1 Variant (Protein) Column (Table 1):

In Table 1, the "N/A" notation in the UBA1 variant (protein) column for a patient in the scRNA-seq cohort requires clarification. The authors should specify whether this indicates the absence of a detected UBA1 variant or missing data. If no amino acid sequence variation was observed in this patient, were additional investigations conducted to clarify why the patient was classified as having VEXAS? Furthermore, if the diagnosis was based on a genetic sequence mutation rather than an amino acid sequence variation, the inclusion of this patient's data (Cohort 16) in the GoT library seems necessary. If there were specific reasons for excluding this data, they should be explicitly explained.

Response: The variant c.118-1G>C is known to be the 4th most common *UBA1* variants in VEXAS patients.⁵ Since the variant is located in a non-coding splicing site (at the boundary of exon 3 and intron 2), the coding protein sequence (p.Met41) is not altered, but the variant disrupt RNA splicing, resulting in aberrant *UBA1* mRNAs that lack the regions around p.Met41 required for translation of the cytoplasmic *UBA1b* isoform.⁶ As the reviewer pointed out, N/A in the UBA1 variant (protein) column may lead to misunderstanding. We have deleted N/A from the cell of VEXAS 16 in the supplementary Table 1. *UBA1* mRNA does not contain the sequence of c.118-1G, and therefore cannot be identified in GoT data

derived from intermediate cDNA sequences, why we excluded VEXAS 16 from GoT library construction. For clarification, we have added the following text in page 5, line 112-115 of the revised manuscript.

“Since GoT was employed to identify *UBA1* mutations within cDNA libraries from VEXAS patients and is not suitable for detection of splice site mutations, GoT library was not constructed from healthy donors or VEXAS 16, who has a *UBA1* mutation at a splicing site (c.118-1G).”

3. Discrepancy Between scRNA-Seq and Flow Cytometry Data (Line 192-201):

The authors identified elevated adaptive-like NK cells in VEXAS patients using scRNA-seq, characterized by high expression of CD3E, LAG3, and IL-32, with no expression of FCER1G and TCRs. However, the observed discrepancy between scRNA-seq and flow cytometry results raises concerns. Specifically, some adaptive-like NK cells lack canonical markers such as KLRC2/NKG2C, which are typically associated with HCMV-induced adaptive NK cells. To address the potential bias introduced by the HCMV-positive VEXAS patient (VEXAS17), it would strengthen the study to include additional experiments using surface markers that are not exclusively linked to HCMV-driven adaptive NK cells. This would help validate the existence of adaptive-like NK cells in VEXAS patients arising from other triggers and confirm their role in the disease.

Response: Based on the reviewer’s suggestion, we collected blood from additional 11 VEXAS patients for flow cytometry analysis. Adaptive-like NK cells were defined as a CD57⁺FCER1G⁻ cell population within CD56^{dim} NK cells, according to previous reports.^{7,8} There was a significant increase in adaptive-like NK cells in VEXAS compared to healthy individuals, consistent with our scRNA-seq data. We now have deleted the original text in page 8, line 196-201 and Supplementary Fig. 5e. Instead, we have changed Supplementary Fig. 5c, d, and added the following text in page 9, line 206-207 of the revised manuscript. In addition, we have added the following new references for adaptive-like NK cells; ref. 53 (Zhang T, Scott JM, Hwang I, Kim S. Cutting edge: antibody-dependent memory-like NK cells distinguished by FcRgamma deficiency. *J Immunol* 190, 1402-1406 (2013)) and ref. 54 (Liu LL, *et al.* Critical Role of CD2 Co-stimulation in Adaptive Natural Killer Cell Responses Revealed in NKG2C-Deficient Humans. *Cell Rep* 15, 1088-1099 (2016))

Original manuscript:

“a finding not reproduced by flow cytometry (Supplementary Fig. 5c, d). Discrepancy between scRNA-seq data and flow cytometry data may be biased due to high frequencies of adaptive-like NK cells from one HCMV⁺ VEXAS patient (VEXAS17) (Supplementary Fig. 5e); as flow cytometry could not distinguish adaptive-like NK cells that arise in response to triggers other than HCMV infection.”

Revised manuscript:

“flow cytometry confirmed the scRNA-seq data (Supplementary Fig. 5c, d)”

New Supplementary Fig. 5c

(Supplementary Fig. 5c) Representative flowcharts for identification of adaptive-like NK cells in peripheral blood. Adaptive-like NK cells were identified as (lowSSC/Singlets/Live/CD56⁺CD3⁻CD14⁻CD19⁻/CD16⁺CD56^{dim}/CD57⁺FcεR1γ⁻).

New Supplementary Fig. 5d

(Supplementary Fig. 5d) Proportions of NK cell subtypes relative to the total number of NK cells in healthy donors (n = 11, blue dots) and VEXAS patients (n = 11, red dots) by flow cytometry. Data are presented as a mean value with SD. P values were calculated using the two sided unpaired Mann-Whitney U test.

4. Clonal Expansion in T Cells (Line 214-247):

The clonal expansion of CD8⁺ T cells with restricted TCR usage is an intriguing finding. However, the lack of shared target antigens or disease-specific TCR clusters raises questions about the drivers of clonal expansion. The authors should discuss whether this phenomenon might reflect non-specific inflammation or neoantigen-driven responses resulting from mtUBA1-related dysregulation in myeloid cells.

Response: Based on scTCR-seq data analysis, we hypothesized that the clonal expansion in CD8⁺ T cells as well as plasmablasts may be triggered by excessive nonspecific antigens produced by mt*UBA1* cells. We have discussed this point in page 14, line 351-353 of the original manuscript (page 15, line 366-368 of the revised manuscript) as shown below.

“Without evidence of disease-specific antigens, nonspecific antigens produced by mt*UBA1* myeloid cells as a result of dysregulated protein degradation may trigger clonal expansion of lymphoid cells in VEXAS.”

5. Broader Implications of Protein Processing in B Cells (Line 280-285):

The significant upregulation of genes related to ER stress and protein processing in plasmablasts suggests enhanced antibody production in VEXAS. Notably, VEXAS patients exhibited increased somatic hypermutation (SHM) in plasmablasts, alongside shifts in immunoglobulin isotype usage, transitioning from IgA dominance observed in healthy donors to IgG predominance in patients. The authors could further explore whether these changes in SHM and immunoglobulin usage contribute to the production of autoantibodies and how they correlate with disease severity and progression.

Response: To our knowledge, there is little evidence that autoantibodies contribute to VEXAS pathophysiology. Our current study is limited in its ability to identify autoreactive B cells. We have discussed autoantibodies in VEXAS in page 14, line 347-350 of the original manuscript (page 15, line 359-365 of the revised manuscript) as shown below.

“Autoantibodies produced by these SHM cells have been associated with development of some autoimmune diseases. An increased ratio of plasmablasts to total B cells in PB occurs in ANCA-associated vasculitis (AAV) and SLE, accompanied by elevated levels of autoantibodies such as anti-dsDNA. Cases of VEXAS associated with SLE and AAV have been reported, and the presence of lupus anticoagulant is not uncommon.”

And

“Screening of autoantibodies in large VEXAS cohorts, in comparison with other autoimmune diseases, would help understand the role of autoreactive B cells in the syndrome.”

6. Lack of Healthy Donor GoT Data and Proportion-Dependent Changes (Supplementary Table 2, Line 111-114, Line 202-212):

In Supplementary Table 2, only GoT data from VEXAS patients are presented. It is unclear whether GoT data were not generated for healthy donor samples. If GoT data were exclusively generated for VEXAS patients, this should be explicitly stated in the manuscript for clarity.

Response: GoT libraries were not constructed from healthy individuals since GoT was employed to identify *UBA1* mutations which do not exist in healthy individuals. We now have added the following description in page 5, line 112-115 of the revised manuscript. Please also see our response to major comment #2.

“Since GoT was employed to identify *UBA1* mutations within cDNA libraries from VEXAS patients and is not suitable for detection of splice site mutations, GoT library was not constructed from healthy donors or VEXAS 16, who has a *UBA1* mutation at a splicing site (c.118-1G).”

Additionally, NK cells are described as being directly impacted by mt*UBA1*, but it would strengthen the study to include data on how the characteristics of NK cells change with varying proportions of mt*UBA1*. If similar proportion-dependent changes are observed in other cell types, those findings should also be included to provide a broader understanding of mt*UBA1*'s impact.

Response: As shown in Fig. 4f, adaptive-like NK cells, which are mostly wild-type *UBA1*, exhibited mild upregulation of inflammatory pathways as compared to other NK cell subtypes, suggesting very low proportion of mt*UBA1* cells in adaptive-like NK cells may result in expression changes in genes involved in inflammatory pathways. To clarify this point, we have plotted several gene module scores along pseudotime ordering, and observed that the module score differences in IFN- γ response, TNF- α response and apoptosis pathways between VEXAS patients and healthy donors became smaller along differentiation, consistent with varying proportion of mt*UBA1* cells among NK cell subtypes. We have created new Supplementary Fig. 5f and added the following text in page 9, line 212-216 of the revised manuscript.

“but differences in gene module scores of those gene pathways between patients and healthy donors decreased with differentiation (Supplementary Fig. 5f), indicating that the upregulation of inflammatory and apoptosis pathways was dependent on the proportion of mt*UBA1* cells in each NK cell subtype.”

New Supplementary Fig. 5f

(Supplementary Fig. 5f) Dynamic changes of gene module scores of the IFN- γ response, TNF- α response and apoptosis pathways in NK cells from VEXAS patients and healthy donors along differentiation. x axis, pseudotime ordering from CD56^{bright} NK cells to adaptive-like NK cells estimated by slingshot; y axis, gene module scores for each pathway.

Lastly, adding plots that depict the ratios of mt*UBA1* to wt*UBA1* across cell types would enhance data visualization and help readers better interpret the results.

Response: As for other cell types, we did not observe varying proportions of mt*UBA1* cells with differentiation; proportion of mt*UBA1* cells in monocyte population did not change with differentiation, and B and T cells were mostly wt*UBA1*. The ratios of mt*UBA1* to wt*UBA1* across major cell types were shown in Fig. 1e of the original manuscript (Fig. 1f of the revised manuscript). To make the figure more explanatory, we have changed text in Fig. 1f as follows:

Original Fig. 1f: mtUBA1 cell ratio

Revised Fig. 1f: mtUBA1 cell ratio in major cell types

Minor comments:

1. Authors should cite the full name of UPR. (Line 73)

Response: Revised accordingly.

2. The naming of DC subtypes alternates between "cDC" and "mDC" throughout the manuscript. It would be preferable to use a consistent term. If "mDC" is chosen, the full name should be specified for clarity (e.g., in Line 142, Line 148, Figure 3a, and Figure 3c).

Response: Revised accordingly.

3. The term "nctMono" appears to be a typo and should be corrected to "ncMono." If this is not a typo, the full name should be provided, and the authors should explain why this new cell type, not mentioned in previous annotations, is being introduced. (Figure 3e)

Response: We have corrected this typographic error.

Supplementary References

1. Venet F, Demaret J, Gossez M, Monneret G. Myeloid cells in sepsis-acquired immunodeficiency. *Ann N Y Acad Sci* **1499**, 3-17 (2021).
2. Saeidi A, *et al.* T-Cell Exhaustion in Chronic Infections: Reversing the State of Exhaustion and Reinvigorating Optimal Protective Immune Responses. *Front Immunol* **9**, 2569 (2018).
3. Stubbins RJ, *et al.* VEXAS syndrome in a female patient with constitutional 45,X (Turner syndrome). *Haematologica* **107**, 1011-1013 (2022).
4. Echerbault R, Bourguiba R, Georgin-Lavialle S, Lavigne C, Ravaiau C, Lacombe V. Comparing clinical features between males and females with VEXAS syndrome: data from literature analysis of patient reports. *Rheumatology (Oxford)* **63**, 2694-2700 (2024).

5. Gutierrez-Rodrigues F, *et al.* Spectrum of clonal hematopoiesis in VEXAS syndrome. *Blood* **142**, 244-259 (2023).
6. Poulter JA, *et al.* Novel somatic mutations in UBA1 as a cause of VEXAS syndrome. *Blood* **137**, 3676-3681 (2021).
7. Zhang T, Scott JM, Hwang I, Kim S. Cutting edge: antibody-dependent memory-like NK cells distinguished by FcRgamma deficiency. *J Immunol* **190**, 1402-1406 (2013).
8. Liu LL, *et al.* Critical Role of CD2 Co-stimulation in Adaptive Natural Killer Cell Responses Revealed in NKG2C-Deficient Humans. *Cell Rep* **15**, 1088-1099 (2016).

RESPONSE TO REVIEWERS' COMMENTS

Reviewer #2 (Remarks to the Author):

The authors have done an outstanding job and thoroughly and convincingly addressed all points raised by the reviewers. I have no further comments.

Response: We thank the reviewer for appreciation of our work and thoughtful comments which helped to improve the manuscript.

Reviewer #3 (Remarks to the Author):

I have carefully reviewed the authors' responses to my initial comments, and I find their revisions and justifications to be appropriate and well-supported. Below is my final assessment of their responses.

1. Gender Bias in Sample Selection

The authors have adequately addressed the concern regarding the exclusion of female patients by referencing relevant literature and explicitly discussing this limitation in the manuscript. Their explanation regarding the rarity of female VEXAS patients due to the X-linked nature of the mutation is well-supported.

2. Clarification of "N/A" Notation in UBA1 Variant Column

The authors have provided a clear explanation regarding the splice-site mutation (c.118-1G>C) and its impact on protein translation, which justifies why "N/A" was originally noted. The removal of "N/A" and the addition of clarifying text improve the manuscript's clarity. Their reasoning for excluding VEXAS16 from the GoT library is scientifically sound and now well-articulated.

3. Discrepancy Between scRNA-Seq and Flow Cytometry Data

The authors have strengthened their findings by conducting additional flow cytometry experiments with an expanded patient cohort. The new data confirms the scRNA-seq findings and eliminates the previous concern regarding a potential bias from a single HCMV-positive patient. The revised Supplementary Figures and added references further support their claims.

4. Clonal Expansion in T Cells

The authors have provided a well-reasoned discussion on how excessive non-specific antigens from mtUBA1 myeloid cells could drive CD8+ T cell clonal expansion. Their revised discussion section appropriately reflects this hypothesis.

5. Broader Implications of Protein Processing in B Cells

While the role of autoantibodies in VEXAS remains uncertain, the authors have adequately contextualized their findings by referencing autoimmune diseases where plasmablast expansion correlates with autoantibody production. Their discussion now highlights the need for future large-cohort studies to explore this potential link.

6. Lack of Healthy Donor GoT Data and Proportion-Dependent Changes

The authors have clarified why GoT data were not generated for healthy donors, making this explicit in the revised manuscript. Additionally, their newly added Supplementary Figure 5f effectively illustrates how inflammatory and apoptotic pathway gene module scores correlate with the proportion of mtUBA1 cells, further strengthening the mechanistic insights of the study. The adjustments to Figure 1f also improve data interpretation.

7. Minor Comments

The manuscript has been revised to address all minor concerns, including terminology inconsistencies, typographical errors, and missing full names of abbreviations. These changes improve the overall clarity and professionalism of the manuscript.

Final Assessment:

The authors have thoroughly and satisfactorily addressed all concerns raised in my initial review. The additional experiments, clarifications, and revisions enhance the robustness and clarity of their findings. I find the manuscript to be well-revised and scientifically sound, and I have no further requests for modification.

Response: We thank the reviewer for appreciation of our revised work and detailed point-by-point comments.